# Learning-Order Autoregressive Models
# with Application to Molecular Graph Generation

Zhe Wang [1 2]   Jiaxin Shi [1]   Nicolas Heess [1]   Arthur Gretton [1 2]   Michalis K. Titsias [1]

## Abstract

Autoregressive models (ARMs) have become the workhorse for sequence generation tasks, since many problems can be modeled as next-token prediction. While there appears to be a natural ordering for text (i.e., left-to-right), for many data types, such as graphs, the canonical ordering is less obvious. To address this problem, we introduce a variant of ARM that generates high-dimensional data using a probabilistic ordering that is sequentially inferred from data. This model incorporates a trainable probability distribution, referred to as an *order-policy*, that dynamically decides the autoregressive order in a state-dependent manner. To train the model, we introduce a variational lower bound on the log-likelihood, which we optimize with stochastic gradient estimation. We demonstrate experimentally that our method can learn meaningful autoregressive orderings in image and graph generation. On the challenging domain of molecular graph generation, we achieve state-of-the-art results on the QM9 and ZINC250k benchmarks, evaluated across key metrics for distribution similarity and drug-likeless.

## 1. Introduction

Artists draw paintings in their own styles stroke by stroke, and chemists synthesize stable molecules step by step following the procedure of chemical reactions. What is common in both cases is that the next data component to be generated is dynamically determined by the current state of the creation process. In machine learning, autoregressive models (ARMs) can encode such sequential dependencies between data dimensions. In general, ARMs represent a high-dimensional data distribution using the product-rule factorization of conditionals. They have become the standard approach for generating text (Brown et al., 2020).

A key limitation of traditional ARMs is their reliance on a predefined, canonical ordering for factorizing the probability distribution over data dimensions. While text data possesses a natural left-to-right ordering, many other data types, such as images and graphs, lack such an obvious or universally applicable ordering. Indeed, the optimal ordering may vary even between individual data points, and the best generation order is often context-dependent. Therefore, it is desirable to derive extensions of ARMs that do not treat the ordering as fixed, but rather as a latent random variable that follows a probability distribution that adapts to the evolving state of the generation process.

One basic way to extend ARMs with probabilistic orderings is any order ARMs (AO-ARMs, Uria et al., 2014), where the ordering follows a uniform distribution over all possible permutations of data dimensions. Such models can be thought of as *order-agnostic*, and also connect with the masked or absorbing discrete diffusion models (Austin et al., 2021; Hoogeboom et al., 2022; Shi et al., 2024; Sahoo et al., 2024; Ou et al., 2024). However, in practice such models are less effective in terms of likelihood scores compared to ARMs (Yang et al., 2019; Hoogeboom et al., 2022). A possible reason is that these models try to solve an extremely challenging training problem, which requires fitting a neural network to adequately match all possible conditional distributions over all orderings of the data dimensions, without learning any preference towards particular orderings.

To address the limitations of fixed order and any order ARMs, we introduce an ARM variant which can flexibly learn probabilistic orderings for generating the data dimensions. Our main contributions and findings include:

- We introduce the Learning-Order Autoregressive Models (LO-ARMs), a novel generative model that learns context-dependent generation orders from data. Specifically, we extend AO-ARMs to incorporate a trainable probability distribution that dynamically decides the sampling order of the data dimensions. We derive a variational lower bound on the exact log-likelihood for

---

[1]Google DeepMind [2]University College London. Correspondence to: Zhe Wang <zhewang@google.com>, Jiaxin Shi <jiaxins@google.com>, Michalis K. Titsias <mtitsias@google.com>.

*Proceedings of the 42nd International Conference on Machine Learning*, Vancouver, Canada. PMLR 267, 2025. Copyright 2025 by the author(s).

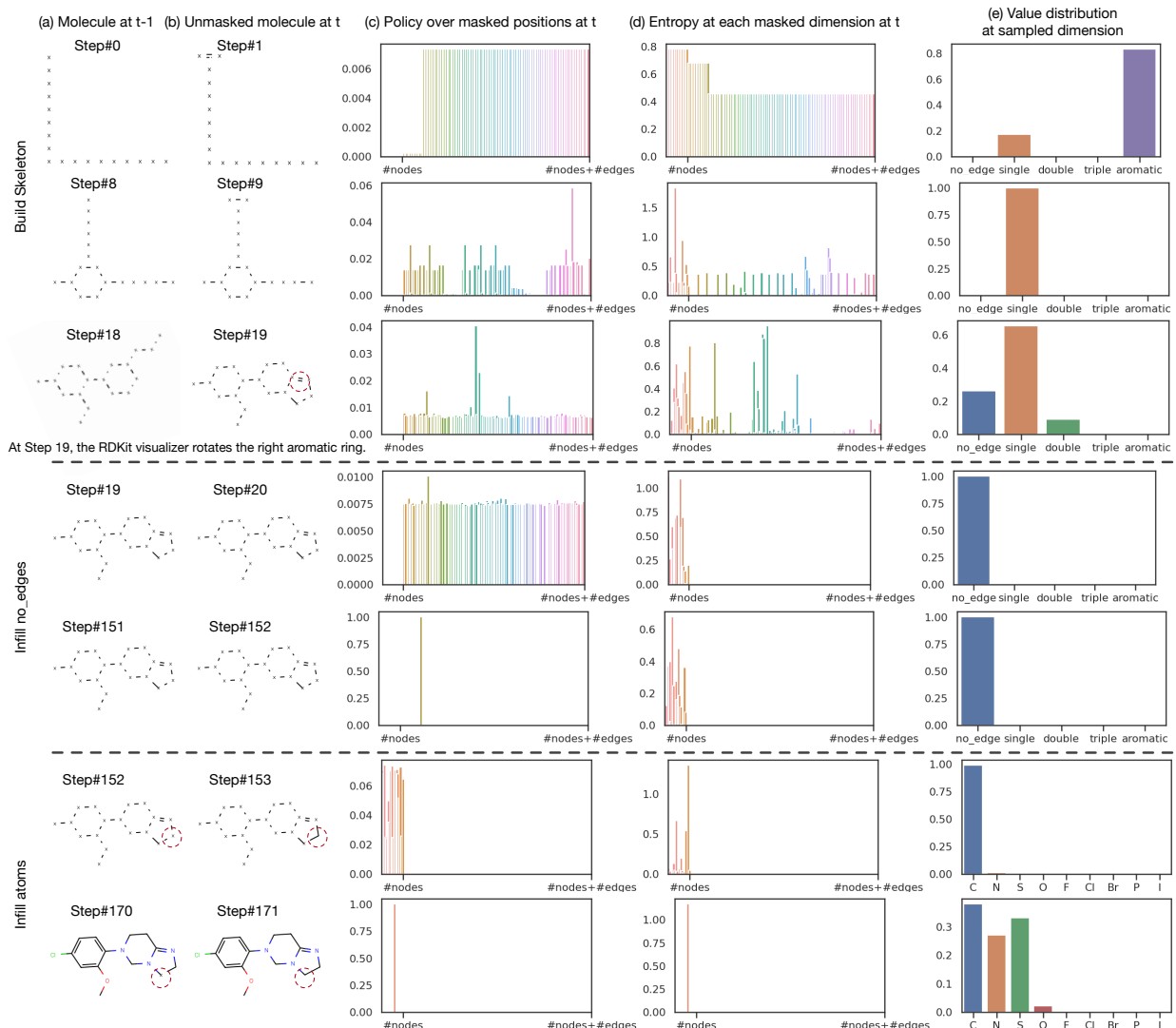

*Figure 1.* **An example of molecule generation with our best LO-ARM model trained on ZINC250k**. Our model generates molecules step-by-step, commencing with all nodes and edges masked (in the figures masked nodes are labeled as ×) and adding one node or edge at a time. First, an *order-policy* selects which dimension (node or edge) to fill, and then a *classifier* determines its value. Each step is illustrated in the provided figures: Columns (a) and (b) illustrate the (partially) generated molecular structures at two successive generation steps. Columns (c), (d), and (e) provide detailed insights: (c) the order-policy's probability distribution over dimensions, (d) the classifier's entropy for each dimension, and (e) the classifier's prediction at the selected dimension. Note that, the order-policy and classifier entropy are zeroed for unmasked dimensions. The generation proceeds through three phases: bond skeleton construction, "no-edge" state population (representing "imaginery" bonds in the dense adjacency matrix), and atom type assignment. In particular, the order-policy learns to favor dimensions with high classifier certainty, demonstrated by an inverse relationship between policy probability and classifier entropy. For example, at Step 1, dimensions with higher probabilities in (c) correspond to lower entropies in (d). This trend is especially evident during the "no-edge" infilling phase, where masked edge dimensions at Step 20 and 152 exhibit high order-policy probabilities and near-zero classifier entropies, while masked node dimensions show the inverse. Importantly, the classifier assigns probabilities to potential values, enabling valid molecular variations. For instance, at Step 171, as shown in (e), the final masked atom has a near-equal chance of being Carbon, Nitrogen, or Sulfur, resulting in three different yet valid molecules. Step 19 also demonstrates this, with each possible edge value offering a different, yet feasible, molecular structure. We provide the full generation path in Appendix B.2.

training the model using stochastic gradient estimation.

- We apply LO-ARMs to two molecule generation tasks (i.e., QM9 and ZINC250k) and obtain state-of-the-

art results for both datasets, as measured by sample quality metrics such as Frèchet ChemNet Distance (FCD) (Preuer et al., 2018), Synthetic Accessibility Score (SAS), Quantitative Estimate of Drug-likeness

(QED), and sample diversity.

- We find that LO-ARMs can learn consistent orderings to generate new molecules with high validity and uniqueness.

- We investigate a variety of architectural choices for LO-ARMs and provide an ablation analysis in the context of molecular graph generation.

## 2. Background

### 2.1. Problem Setup

We assume data vectors $\boldsymbol{x} = (x_1, \ldots, x_L)$ where each dimension $x_i$ takes values from a set $\mathcal{X}$, which could be real or discrete. Without loss of generality, we assume $\boldsymbol{x}$ is a vector of $L$ categorical variables, so that $\mathcal{X}$ is a discrete set of $m = |\mathcal{X}|$ categories. For example, for molecular graphs, $\mathcal{X} = \mathcal{V} \bigcup \mathcal{E}$, where $\mathcal{V}$ and $\mathcal{E}$ are the atom and bond types respectively. An autoregressive model defines a joint probability distribution over $\boldsymbol{x}$ that factorizes as

$$p_\theta(\boldsymbol{x}) = \prod_{i=1}^{L} p_\theta(x_i|\boldsymbol{x}_{<i}), \qquad (1)$$

where $x_i$ denotes the $i$-th dimension of $\boldsymbol{x}$, $\boldsymbol{x}_{<i} = (x_1, \ldots, x_{i-1})$ denotes the first $i - 1$ elements of the vector $\boldsymbol{x}$ and $p_\theta(x_i|\boldsymbol{x}_{<i})$ is the conditional distribution with the convention $p_\theta(x_1|\boldsymbol{x}_{<1}) = p_\theta(x_1)$.

Sampling from the model generates the data dimensions sequentially starting from $x_1$ and ending at $x_L$. As mentioned in the introduction, having a fixed or pre-specified order can often be disadvantageous, since it may introduce inappropriate inductive bias when modeling data without a natural ordering. AO-ARMs (Uria et al., 2014; Hoogeboom et al., 2022) address this problem by training a model that can generate the data dimensions under any random ordering drawn uniformly from $L!$ permutations of the indices $\{1, \ldots, L\}$. Given a permutation $\sigma$ the model joint distribution factorizes as

$$p_\theta(\boldsymbol{x}|\sigma) = \prod_{i=1}^{L} p_\theta(x_{\sigma_i}|\boldsymbol{x}_{\sigma_{<i}}), \qquad (2)$$

where $\sigma_{<i}$ denotes the indices of the $i - 1$ first elements under the permutation $\sigma$. If $p(\sigma)$ denotes the uniform distribution over the $L!$ permutations, the parameters $\theta$ are found by maximizing the expected log-likelihood (per data point):

$$\mathbb{E}_{p(\sigma)}\left[\log p_\theta(\boldsymbol{x}|\sigma)\right]. \qquad (3)$$

As noted by Hoogeboom et al. (2022), a way to interpret the above objective is as a variational lower bound on the log-likelihood of a probabilistic latent variable model. Specifically, if $\sigma$ is the latent variable corresponding to the data

point $\boldsymbol{x}$ (so at training for each example $\boldsymbol{x}^{(n)}$ there is a different $\sigma^{(n)}$), the log-likelihood can be lower bounded as $\log p_\theta(\boldsymbol{x}) = \log \sum_\sigma p(\sigma)p(\boldsymbol{x}|\sigma) \geq \mathbb{E}_{p(\sigma)}[\log p(\boldsymbol{x}|\sigma)]$, which yields the training objective in Equation (3). Note that in practice this objective is optimized stochastically; see Uria et al. (2014) for more details.

### 2.2. Autoregressive Generation as Unmasking Process

Here, we describe a convenient way to characterize the sequential generation process for discrete data which we will use for building our proposed method in Section 3.

Given that each discrete data dimension (or *token*) $x_i, i \in \{1, \ldots, L\}$ takes $m$ categorical values we further augment the space with an extra *auxiliary* category or mask. Thus, we represent each $x_i$ as an $m + 1$-dimensional (rather than $m$-dimensional) one-hot vector where the final $m + 1$-th value indicates that $x_i$ is masked. Then we can model the generation process with ARMs as an unmasking process. Specifically, starting with a fully masked state $\bar{\boldsymbol{x}} = (\bar{x}_1, \ldots, \bar{x}_L)$, where $\bar{x}_i$ represents the mask, at each step, we choose a dimension $\bar{x}_i$ and "unmask it" which means to sample a categorical value $x_i$ among the $m$ categories. Then we repeat this process until all dimensions are unmasked, which yields a final generated data point $\boldsymbol{x}$. Under this representation our model also connects with recent masked discrete diffusion models (see, e.g., Shi et al., 2024) as we further discuss in Related Work. However, note that unlike discrete diffusion models we do not specify a forward process in our framework, but only the unmasking or backward process.

## 3. Learning Order ARM

We replace the uniform prior distribution $p(\sigma)$ in AO-ARMs by with a learnable distribution over orderings. We call this distribution the *order-policy* since it dynamically decides the next dimension to generate by conditioning on the already generated data dimensions of $\boldsymbol{x}$. Next, we focus on the general description of our method and the key design choices, and delay the technical details of specifying it to practical tasks, including images (Appendix A) and molecular graphs (Appendix E) generation. Specifically, this section unfolds as follows. We introduce the order-policy in Section 3.1, define the learning objective in Section 3.2, and then describe options for the model parametrizations and neural network architectures in Section 3.3.

### 3.1. Model with Order-Policy

Sampling an ordering $\sigma$ can be represented by a set of $L$ latent variables $z_i, i = 1, \ldots, L$, so that

$$p(\boldsymbol{z}) = \prod_{i=1}^{L} p(z_i|\boldsymbol{z}_{<i}), \qquad (4)$$

where $z_1 \sim p(z_1) = p(z_1|\mathbf{z}_{<1})$ is a categorical variable that takes $L$ values from the set $\{1, \ldots, L\}$, and each subsequent $z_i \sim p(z_i|\mathbf{z}_{<i})$ takes $L - i + 1$ values from the set $\mathbf{z}_{\geq i} = \{1, \ldots, L\} \setminus \mathbf{z}_{<i}$, where $\mathbf{z}_{<i} = (z_1, \ldots, z_{i-1})$. The probability distribution $p_\theta(\boldsymbol{x})$ can be written as

$$p_\theta(\boldsymbol{x}) = \sum_{\boldsymbol{z}} p(\boldsymbol{z})p_\theta(\boldsymbol{x}|\boldsymbol{z}) = \sum_{\boldsymbol{z}} \prod_{i=1}^{L} p(z_i|\mathbf{z}_{<i})p_\theta(x_{z_i}|\boldsymbol{x}_{\mathbf{z}_{<i}}),$$

where the two conditionals $p(z_i|\mathbf{z}_{<i})$ and $p_\theta(x_{z_i}|\boldsymbol{x}_{\mathbf{z}_{<i}})$ follow side-by-side an autoregressive structure that unfolds from $i = 1$ to $i = L$. When each $p(z_i|\mathbf{z}_{<i})$ is uniform then $p(\boldsymbol{z})$ is an autoregressive representation of the uniform distribution over the $L!$ permutations. In this case, the model reduces to standard AO-ARMs.

In our proposed method, named learning order ARM (LO-ARM), we use the latent variables $\boldsymbol{z}$ but we replace $p(z_i|\mathbf{z}_{<i})$ with a more informed distribution. We call this distribution *order-policy* and we define it as follows.

**Definition 3.1.** The order-policy is a distribution over $\boldsymbol{z}$ that follows the factorization

$$p_\theta^x(\boldsymbol{z}) = \prod_{i=1}^{L} p_\theta(z_i|\mathbf{z}_{<i}, \boldsymbol{x}_{\mathbf{z}_{<i}}), \tag{5}$$

where each factor $p_\theta(z_i|\mathbf{z}_{<i}, \boldsymbol{x}_{\mathbf{z}_{<i}})$ is a parameterized categorical distribution over the index $z_i$ of the next data dimension in the autoregressive order that conditions not only on the indices $\mathbf{z}_{<i}$ but also on the corresponding data dimensions $\boldsymbol{x}_{\mathbf{z}_{<i}} = (x_{z_1}, \ldots, x_{z_{i-1}})$.

The order-policy enables modeling of the sequential dependence structure of data dimensions that can exist in the true data distribution. Therefore, it is desirable to flexibly learn this distribution from data. A full description of a specific parametrization of the order-policy and an overall model architecture are given in Section 3.3. Before proceeding to this, we discuss a general training procedure in the next section, based on variational inference and stochastic gradient estimation.

### 3.2. Training with Variational Inference

To train the LO-ARM model from a set of training examples $\mathcal{D} = \{\boldsymbol{x}^{(n)}\}_{n=1}^{N}$ we want to maximize the log-likelihood

$$\sum_{n=1}^{N} \log p_\theta(\boldsymbol{x}^{(n)}) = \sum_{n=1}^{N} \log \sum_{\boldsymbol{z}^{(n)}} p_\theta(\boldsymbol{z}^{(n)}, \boldsymbol{x}^{(n)}).$$

For simplicity, we only consider one data point for now and drop index $n$. The joint distribution $p_\theta(\boldsymbol{z}, \boldsymbol{x})$ can then be factorized as

$$p_\theta(\boldsymbol{z}, \boldsymbol{x}) = \prod_{i=1}^{L} p_\theta(z_i|\mathbf{z}_{<i}, \boldsymbol{x}_{\mathbf{z}_{<i}})p_\theta(x_{z_i}|\boldsymbol{x}_{\mathbf{z}_{<i}}), \tag{6}$$

where the factors $p_\theta(z_i|\mathbf{z}_{<i}, \boldsymbol{x}_{\mathbf{z}_{<i}})$ and $p_\theta(x_{z_i}|\boldsymbol{x}_{\mathbf{z}_{<i}})$ depend on parameters $\theta$ that we want to learn. Since the exact likelihood is intractable, we will maximize an evidence lower bound (ELBO) on the log-likelihood. We use an amortized variational distribution over $\boldsymbol{z}$ that conditions on the full data vector $\boldsymbol{x}$, and has the general form

$$q_\theta(\boldsymbol{z}|\boldsymbol{x}) = \prod_{i=1}^{L} q_\theta(z_i|\mathbf{z}_{<i}, \boldsymbol{x}), \tag{7}$$

where the specific parametrized form of $q_\theta$ is given in Equation (12) and Section 3.3. Structurally, each variational factor $q_\theta(z_i|\mathbf{z}_{<i}, \boldsymbol{x})$ has a similar form as the order-policy factor $p_\theta(z_i|\mathbf{z}_{<i}, \boldsymbol{x}_{\mathbf{z}_{<i}})$, but the difference is that the former is allowed to condition on the full $\boldsymbol{x}$ while the latter only on $\boldsymbol{x}_{\mathbf{z}_{<i}}$. This amortized factorization is one of our key design considerations, and makes model training practically feasible while still allowing us to efficiently compute an unbiased estimate of the ELBO. We will discuss this point at the end of the section.

Using $q_\theta$ we can lower bound the log likelihood as follows:

$$\log p_\theta(\boldsymbol{x}) \geq \sum_{\boldsymbol{z}} q_\theta(\boldsymbol{z}|\boldsymbol{x}) \log \frac{p_\theta(\boldsymbol{z}, \boldsymbol{x})}{q_\theta(\boldsymbol{z}|\boldsymbol{x})} =$$

$$\sum_{\boldsymbol{z}} q_\theta(\boldsymbol{z}|\boldsymbol{x}) \sum_{i=1}^{L} \log \frac{p_\theta(z_i|\mathbf{z}_{<i}, \boldsymbol{x}_{\mathbf{z}_{<i}})p_\theta(x_{z_i}|\boldsymbol{x}_{\mathbf{z}_{<i}})}{q_\theta(z_i|\mathbf{z}_{<i}, \boldsymbol{x})} =$$

$$\sum_{i=1}^{L} \mathbb{E}_{q_\theta(\mathbf{z}_{<i}|\boldsymbol{x})} \left[ \mathbb{E}_{q_\theta(z_i|\mathbf{z}_{<i}, \boldsymbol{x})} \left[ \log \frac{p_\theta(z_i|\mathbf{z}_{<i}, \boldsymbol{x}_{\mathbf{z}_{<i}})p_\theta(x_{z_i}|\boldsymbol{x}_{\mathbf{z}_{<i}})}{q_\theta(z_i|\mathbf{z}_{<i}, \boldsymbol{x})} \right] \right]$$

$$= \sum_{i=1}^{L} \mathbb{E}_{q_\theta(\mathbf{z}_{<i}|\boldsymbol{x})} \left[ F_\theta(\mathbf{z}_{<i}, \boldsymbol{x}) \right]. \tag{8}$$

To obtain the final expression, we performed the exact expectation over $z_i \in \mathbf{z}_{\geq i} = \{1, \ldots, L\} \setminus \mathbf{z}_{<i}$ (i.e., taking values over the set of all currently masked dimensions) and defined the function $F_\theta(\mathbf{z}_{<i}, \boldsymbol{x})$ as shorthand for the long expectation highlighted in blue. We also analytically marginalized out all future latent variables $\mathbf{z}_{>i} = \{1, \ldots, L\} \setminus \mathbf{z}_{\leq i}$ since the function $F_\theta(\mathbf{z}_{<i}, \boldsymbol{x})$ does not depend on these. Computing the full ELBO is too expensive, and thus in practice we construct an unbiased estimate by sampling one term in the sum $\sum_{i=1}^{L}$ together with one $\mathbf{z}_{<i} \sim q_\theta(\mathbf{z}_{<i}|\boldsymbol{x})$. Thus we obtain the stochastic estimate of our loss as

$$\mathcal{L}(\theta) = -LF_\theta(\mathbf{z}_{<i}, \boldsymbol{x}), \quad \mathbf{z}_{<i} \sim q_\theta(\mathbf{z}_{<i}|\boldsymbol{x}). \tag{9}$$

For the special case where each variational factor $q_\theta(z_i|\mathbf{z}_{<i}, \boldsymbol{x})$ and $p_\theta(z_i|\mathbf{z}_{<i}, \boldsymbol{x}_{\mathbf{z}_{<i}})$ are non-learnable and set to uniform distributions, the stochastic negative ELBO in Equation (9) reduces to

$$\mathcal{L}(\theta) = -\frac{L}{L - i + 1} \sum_{z_i \in \mathbf{z}_{\geq i}} \log p_\theta(x_{z_i}|\boldsymbol{x}_{\mathbf{z}_{<i}}), \tag{10}$$

**Algorithm 1** Training with LO-ARM

Given a dataset $\mathcal{D}$, $p_\theta(\cdot|\boldsymbol{x}_{\boldsymbol{z}_{<i}})$, $p_\theta(\cdot|\boldsymbol{z}_{<i}, \boldsymbol{x}_{\boldsymbol{z}_{<i}})$, $q_\theta(\cdot|\boldsymbol{x})$
**while** training **do**
    Uniformly sample $\boldsymbol{x} = (x_1, \ldots, x_L)$ from $\mathcal{D}$
    Sample $\boldsymbol{z}^1 \sim q_\theta(\cdot|\boldsymbol{x})$ and $\boldsymbol{z}^2 \sim q_\theta(\cdot|\boldsymbol{x})$
    Sample $i \sim \text{Uniform}[1, \ldots, L]$
    Set $\boldsymbol{z}^1_{<i} = (z^1_1, \ldots, z^1_{i-1})$ and $\boldsymbol{z}^2_{<i} = (z^1_1, \ldots, z^2_{i-1})$
    Get $\boldsymbol{x}_{\boldsymbol{z}^1_{<i}}, \boldsymbol{x}_{\boldsymbol{z}^2_{<i}}$ by masking dimensions at $\boldsymbol{z}^1_{\geq i}, \boldsymbol{z}^2_{\geq i}$
    Compute RLOO unbiased gradient from (11) and update $\theta$ via SGD
**end while**

---

**Algorithm 2** Unconditional sampling from LO-ARM

Initialize a fully masked state $\bar{\boldsymbol{x}} = (\bar{x}_1, \ldots, \bar{x}_L)$
Initialize $\bar{\boldsymbol{z}} = \{1, \ldots, L\}$ as the indices to be sampled
**for** step $i = 1, 2, \ldots, L$ **do**
    Sample data index $z_i \sim p_\theta(\cdot|\boldsymbol{z}_{<i}, \boldsymbol{x}_{\boldsymbol{z}_{<i}})$
    Sample data dimension value $\hat{x}_{z_i} \sim p_\theta(\cdot|\boldsymbol{x}_{\boldsymbol{z}_{<i}})$
    Unmask $\bar{x}_{z_i}$ with $\hat{x}_{z_i}$: $\bar{\boldsymbol{x}} \leftarrow \bar{\boldsymbol{x}} \setminus \bar{x}_{z_i} \cup \{\hat{x}_{z_i}\}$
    Remove $z_i$ from $\bar{\boldsymbol{z}}$: $\bar{\boldsymbol{z}} \leftarrow \bar{\boldsymbol{z}} \setminus z_i$
**end for**
Return $\bar{\boldsymbol{x}}$

---

which is precisely the stochastic objective used for training AO-ARMs; see Equation (12) in Uria et al. (2014). In other words, the objective in Equation (9) generalizes these previous objectives.

Unlike AO-ARMs, however, the stochastic objective in Equation (9) is more complex, which therefore would have higher variance, In order to propagate gradients through the sampling step from the discrete distribution $q_\theta$ and reduce variance, we also need REINFORCE gradient estimation (see, e.g., Shi et al., 2022). To keep it simple, we use the REINFORCE leave-one-out (RLOO) estimator (Salimans & Knowles, 2014; Kool et al., 2019a) with two samples. First we select a single random index $i \in \{1, \ldots, L\}$. Then given this fixed $i$ we draw two sample paths of previous indices $\boldsymbol{z}^1_{<i}$ and $\boldsymbol{z}^2_{<i}$, by sampling from $q_\theta(\boldsymbol{z}_{<i}|\boldsymbol{x})$. The unbiased gradient over $\theta$ is then written as

$$\frac{L}{2}\{\left(\nabla_\theta \log q_\theta(\boldsymbol{z}^1_{<i}|\boldsymbol{x}) - \nabla_\theta \log q_\theta(\boldsymbol{z}^2_{<i}|\boldsymbol{x})\right)\Delta F$$
$$+ \nabla_\theta F_\theta(\boldsymbol{z}^1_{<i}, \boldsymbol{x}) + \nabla_\theta F_\theta(\boldsymbol{z}^2_{<i}, \boldsymbol{x})\}, \qquad (11)$$

where $\Delta F = F_\theta(\boldsymbol{z}^1_{<i}, \boldsymbol{x}) - F_\theta(\boldsymbol{z}^2_{<i}, \boldsymbol{x})$.

Algorithm 1 outlines the whole training procedure. In particular, $\boldsymbol{z} = (z_1, \ldots, z_L)$ is a full permutation of $\{1, \ldots, L\}$ and fast sampling of $\boldsymbol{z}$, e.g., by requiring only a single NN forward pass, is critical to obtain scalable training. To enable this, we construct the full $q_\theta$ as an amortized (by data vector $\boldsymbol{x}$) Plackett-Luce model (Plackett, 1975). More precisely, we assume the vector of logits $g_\theta(\boldsymbol{x}) \in \mathbb{R}^L$, which

is a non-linear function that receives as input the data $\boldsymbol{x}$. Based on these logits each variational factor has the form

$$q_\theta(z_i = k|\boldsymbol{z}_{<i}, \boldsymbol{x}) = \frac{e^{g_{\theta,k}(\boldsymbol{x})}}{\sum_{k' \in \boldsymbol{z}_{\geq i}} e^{g_{\theta,k'}(\boldsymbol{x})}}. \qquad (12)$$

The logits are combined in an autoregressive way based on the factorization in Equation (7). Computing and sampling from the variational distribution is very fast, since it requires a single evaluation to obtain and store the vector of logits $g_\theta(\boldsymbol{x})$, and sampling from $q_\theta$ in (7), with factors given by (12), has negligible additional cost. In fact a full path $\boldsymbol{z}$ can be sampled at once in parallel using the Gumbel-top-k trick (Kool et al., 2019b), as we do in our implementation. The exact parametrization of $g_\theta(\boldsymbol{x})$ using the neural architecture is discussed next in Section 3.3. Moreover, similar to training variational autoencoders (Kingma & Welling, 2014), we can obtain unbiased gradients of the ELBO by sampling the discrete latent variable $\boldsymbol{z} \sim q_\theta$, which allows for training $q_\theta$ and the model $p_\theta$ jointly with a single optimizer.

Once training is completed, we can generate new samples from the model using Algorithm 2. Specifically, we can generate a new sample $\hat{\boldsymbol{x}} = (\hat{x}_1, \ldots, \hat{x}_L)$ autoregressively starting from a fully masked state $\bar{\boldsymbol{x}} = (\bar{x}_1, \ldots, \bar{x}_L)$. To do this, at step $i \in \{1, \ldots, L\}$, we unmask $\bar{x}_{z_i}$ to $\hat{x}_{z_i}$, which is sampled with the categorical distribution. Note that we only need the model order-policy over $z_i \sim p_\theta(\cdot|\boldsymbol{z}_{<i}, \boldsymbol{x}_{\boldsymbol{z}_{<i}})$ and the categorical distribution $\hat{x}_{z_i} \sim p_\theta(\cdot|\boldsymbol{x}_{\boldsymbol{z}_{<i}})$ for inference.

### 3.3. Parametrization of the Distributions

Here, we discuss specific parametric forms for the model and variational distributions. Since $x_{z_i}$ is discrete, we parameterize the model conditionals $p_\theta(x_{z_i=k}|\boldsymbol{x}_{\boldsymbol{z}_{<i}})$ for $k = 1, \ldots, L$ as $L$ *classifiers* with a Neural Network (NN) having $L$ heads,

$$p_\theta(x_{z_i=k}|\boldsymbol{x}_{\boldsymbol{z}_{<i}}) = \text{softmax}(f_{\theta,k}(\bar{\boldsymbol{x}}_{\boldsymbol{z}_{<i}})), \qquad (13)$$

where $\bar{\boldsymbol{x}}_{\boldsymbol{z}_{<i}}$ is the state with the $\boldsymbol{z}_{<i}$ dimensions being unmasked and the rest masked, while $f_{\theta,k}(\bar{\boldsymbol{x}}_{\boldsymbol{z}_{<i}}) \in \mathbb{R}^m$ are the logits of the $k$-th softmax head over $m$ classes. The specific architecture of these classifiers could be task dependent. For instance, we employ UNet (Ronneberger et al., 2015) for image generation (Appendix A) and Graph Transformer (Vignac et al., 2023) for graph generation (see Appendix E).

Next, we parameterize the model order-policy factors $p_\theta(z_i = k|\boldsymbol{z}_{<i}, \boldsymbol{x}_{\boldsymbol{z}_{<i}})$ using $L$ functions $h_{\theta,k}(\cdot) \in \mathbb{R}$, $k = 1, \ldots, L$ so that

$$p_\theta(z_i = k|\boldsymbol{z}_{<i}, \boldsymbol{x}_{\boldsymbol{z}_{<i}}) = \frac{e^{h_{\theta,k}(\bar{\boldsymbol{x}}_{\boldsymbol{z}_{<i}})}}{\sum_{k' \in \boldsymbol{z}_{\geq i}} e^{h_{\theta,k'}(\bar{\boldsymbol{x}}_{\boldsymbol{z}_{<i}})}}. \qquad (14)$$

We explore two options to define $h_{\theta,k}(\cdot)$: i) An **entropy-based** parametrization, where these functions are scaled

entropies of $p_\theta(x_k|\boldsymbol{x}_{\boldsymbol{z}_{<i}})$ (computed from logits $f_{\theta,k}$ discussed above). This can favor sampling dimensions $z_i$ with higher certainty (about the value of $x_{z_i}$) early on. ii) A **shared-torso** parametrization, where we add an extra final linear layer to the $f_\theta$ outputting $L$ scalars providing the $h_{\theta,k}(\cdot)$ values.

Finally, we have already described the general form of the variational factors $q_\theta(z_i = k|\boldsymbol{z}_{<i},\boldsymbol{x})$ in Equation (12), so what remains is to specify the vector of logits $g_\theta(\boldsymbol{x})$. We explore two parametrizations: 1) A **shared-torso** parameterization where we add another final linear layer to $f_\theta$ (similarly to (ii) above) outputting $L$ values for $g_\theta(\boldsymbol{x})$. 2) A **separate** NN with $L$ outputs to model $g_\theta(\boldsymbol{x})$. The motivation of this latter option is to more freely capture the form of the variational distribution.

We provide further details on the above options in Appendix F. In the experiments we present an ablation for these options in the context of molecule graph generation.

## 4. Related Work

**Learning Non-Monotonic Autoregressive Orderings** has been studied extensively in recent literature (e.g., Li et al., 2021; Gu et al., 2019; Welleck et al., 2019), and is challenged by the need to find an optimal permutation from a factorial ($L!$) search space, where $L$ is the sequence length. Some methods reduce this space with domain assumptions (Welleck et al., 2019; Gu et al., 2019). Specifically, Welleck et al. (2019) proposes to use a tree-based recursive generation method to learn abitrary generation orders, and Gu et al. (2019) combines 1) pretaining with prescribed base orderings and 2) fine-tuning those orderings with Searched Adaptive Order (SAO). More relevant to our work, Variational Order Inference (VOI) (Li et al., 2021) learns orderings directly from data using a policy gradients procedure, but it needs to optimize a complex variational ordering distribution which has an intractable normalizing constant and requires a Bethe-type approximation. In contrast, the variational distribution in our LO-ARM is fully tractable, and it allows us to use fast and exact unbiased gradient-based optimization of the ELBO using REINFORCE leave-one-out.

**Orders of autogressive graph generation.** Unlike image and text domains, the generation order of ARMs has been a central focus in the graph domain. Early work on graph generative models (Li et al., 2018) mentioned the difficulty of learning the ordering and eventually settled on using either uniform or fixed ordering. You et al. (2018) introduced a fixed BFS ordering scheme. Chen et al. (2021); Kong et al. (2023) extended AO-ARMs to enable a partially learnable ordering over nodes, where the edges connecting a newly added node to existing nodes are generated immediately following the new node. Bu et al. (2023) reformulated the node ordering problem as a dimensionality reduction. These existing works mostly follow a paradigm of incrementally adding new nodes and connecting them to existing nodes. By contrast, our work provides a general order learning framework for ARMs, which enables more flexible ordering relationships between nodes and edges.

**Discrete diffusion and application to graph generation.** Our method also relates to discrete diffusion models based on absorbing or masked diffusion (Austin et al., 2021; Lou et al., 2024; Shi et al., 2024; Sahoo et al., 2024; Ou et al., 2024). Similar to masked diffusion formulations, our specific neural architecture for discrete data assumes that all not-yet-generated data dimensions are assigned the mask category. One important difference with masked diffusion is that we learn a non-uniform and data-dependent order of the data dimensions in a flexible way, using a NN parametrized order-policy or a confidence-based (using entropic uncertainties) policy. In contrast, masked diffusion methods operate similarly to AO-ARMs (Hoogeboom et al., 2022), i.e. they generate discrete tokens using a completely random order. Further, unlike diffusion-based methods, our approach does not explicitly specify a forward noising process. Instead, we only define a *backward generative model* which generates samples from a fully masked state, and our model learns a variational order distribution ($q_\theta$) from data to encode the token unmasking sequence. Analogous to discrete Variational Autoencoders (VAEs), $q_\theta$ is optimized jointly with the generative model using a REINFORCE-style gradient estimator. This contrasts with diffusion models like DiGress (Vignac et al., 2023), which rely on a fixed forward process instead of a learned variational distribution.

## 5. Evaluation and Analysis

We are interested in two questions: 1) whether LO-ARM can learn meaningful orderings to generate data, and 2) whether the learned orderings can yield better generation performance. To answer these, we apply LO-ARMs to two tasks: 1) a toy image generation task on the MNIST dataset to showcase the ordering the model learns, and 2) molecular graph generation on the QM9 and ZINC250K datasets (Xu et al., 2019; Dwivedi et al., 2023). In both tasks, the data does not exhibit a "canonical" ordering. For the MNIST task, we provide a qualitative illustration that LO-ARM prefers to sample first the border pixel values and then the "digit" pixels in the centered foreground. Presumably, the reason is that the model learns to prefer an order where the "easier" border pixels are generated first, which would be of higher certainty. We provide an illustrative figure together with a more detailed description in Appendix A. In the present section, we focus on the quantitative evaluation of LO-ARM on molecular graph generation tasks.

*Table 1.* Molecule generation on QM9. We directly cite the results of other methods. The negative loglikelihoods (NLLs) of our methods are evaluated against the test set, and other metrics are calculated with the generated samples with the corresponding methods.

| Method | | | NLL↓ | Validity%↑ | Uniqueness%↑ | FCD↓ |
|---|---|---|---|---|---|---|
| Ground truth (test data) | | | - | 99.3 | 100 | 0.005 |
| GDSS | | | - | 95.7 | 98.50 | 2.900 |
| DiGress | | | 69.6 | 99.00 | 96.20 | - |
| GraphARM | | | - | 90.25 | 95.62 | 1.22 |
| CatFlow | | | - | 99.81 | **99.95** | 0.441 |
| Our Results | $p_\theta$ | $q_\theta$ | | | | |
| AO-ARM | | | $\leq 24.66$ | 98.88 | 99.11 | 0.671 |
| LO-ARM-st-st | shared torso | shared torso | $\leq 22.38$ | 99.05 | 98.59 | 0.437 |
| LO-ARM-st-sep | shared torso | separate | $\leq 21.42$ | **99.85** | 98.85 | **0.240** |

We evaluate LO-ARM on two widely-adopted molecular graph generation benchmasks: QM9 and ZINC250K (Xu et al., 2019; Dwivedi et al., 2023). We follow the standard setup - e.g., in Eichelsbacher & Reinert (2008); Vignac et al. (2023); Jo et al. (2022), including data preprocessing, network parametrization, and evaluation metrics. We represent molecules as graphs with dense adjacency matrices (see Appendix E). For each model variant, we generate 16384 samples and evaluate them according to two aspects: 1) validity and uniqueness of individual molecules and 2) Frèchet ChemNet Distance (FCD), which evaluates the distance between the distributions of true and generated molecules using the activations of ChemNet (Preuer et al., 2018). Note that, these two classes of metrics are not explicitly correlated, since invalid molecules may have latent activations close to valid ones. As we can see in Table 2, the validity of methods with similar FCDs may vary in a big range (e.g., GraphAF, GraphARM, EDP-GNN and SPECTRE). On the other hand, if the true dataset has high quality, we could still expect molecule validity to be improved when just optimizing FCD-related objectives, e.g., the lower bound of log-likelihoods. In this study, we focus more on the evaluation with FCD to show the generative capability of LO-ARM, and we will show how our validity and uniqueness correlate with FCD.

To see the effect of the order policy, we introduce two baselines, i.e., 1) AO-ARM in which both the variational ($q_\theta$) and model ($p_\theta$) order policies are set to uniform, 2) Biased-AO-ARM (in Table 2), in which the order policies always uniformly samples edges first and then uniformly samples nodes after all edges are unmasked. In addition, we fix the parameterization of the classifier $f_{\theta,z_i}$ to be the same as two recent state-of-the-arts DiGress (Vignac et al., 2023) and CatFlow (Eijkelboom et al., 2024), and ablate different combinations of the parametrizations of model $h_{\theta,k}$ and variational policy distributions. These parameterizations are described in Section 3.3 and in Appendix F with more details. The training configurations, including network ar-

chitectures, are described in Appendix E. The results are summarised in Table 1 and Table 2, and samples generated with our best models are shown in Appendix B.1.

First, on QM9 (Table 1), we can see that all variants of LO-ARM consistently outperform AO-ARM. With similar model capacity, LO-ARM-st-st (both $p_\theta$ and $q_\theta$ share torso with the classifier) achieves similar performance as the previous best method CatFlow. Moreover, with increased capacity for the variational order policy, LO-ARM-st-sep (shared torso for $p_\theta$ and separate NN for $q_\theta$) achieves new state-of-the-art results in FCD, as well as competitive validity and uniqueness. Note that the cost of inference with LO-ARM-st-sep remains close to other Graph Transformer-based methods, because only the posterior model order policy ($p_\theta$) and the classifier are used during inference.

Next, we evaluate LO-ARM with ZINC250k (Table 2), which is a much more challenging dataset made of larger drug-like molecules, as data dimensions increase in $\mathcal{O}(n^2)$ when representing $n$-atom molecules as graphs. Comparing LO-ARM with our own baselines (i.e., AO-ARM and Biased-AO-ARM), we see that introducing order policies improves the performance with a large margin of gain on all metrics (especially validity). Furthermore, our model demonstrates substantial FCD improvement, indicating higher sample quality than previous molecule generative models in terms of resembling the true data distribution.

We observe that there is generally a gap between the validity of ARMs and that of the best method on this task. We hypothesize that this is caused by the errors accumulated during autoregressive sampling, as we unmask a new dimension at a time without refining the previously unmasked. Still, as demonstrated by the gain in validity compared to AO-ARM and GraphARM, LO-ARM has largely fixed this problem by training an order-policy that maximizes the ELBO. Moreover, this accumulated error can be further reduced by employing Top-$p$ sampling (Holtzman et al.), a popular method used in autoregressive language models to

*Table 2.* Molecule generation on ZINC250K. In particular, the NLLs of the LO-ARMs with the Top-$p$ sampling are omitted as the sampler is only used in inference time and does not change the training-time metrics.

| Method | | | | NLL↓ | Validity%↑ | Uniqueness%↑ | FCD↓ |
|---|---|---|---|---|---|---|---|
| Ground truth (test data) | | | | - | 100.00 | 99.88 | 0.005 |
| GraphDF | | | | - | 90.61 | 99.63 | 33.55 |
| MoFlow | | | | - | 63.11 | 99.99 | 20.93 |
| SPECTRE | | | | - | 90.20 | 67.05 | 18.44 |
| EDP-GNN | | | | - | 82.97 | 99.79 | 16.74 |
| GraphARM | | | | - | 88.23 | 99.46 | 16.26 |
| GraphAF | | | | - | 68.47 | 98.64 | 16.02 |
| GDSS | | | | - | 97.01 | 99.64 | 14.66 |
| CatFlow | | | | - | **99.21** | **100.00** | 13.21 |
| Our Results | $p_\theta$ | $q_\theta$ | Top-$p$ | | | | |
| AO-ARM | | | 1.0 | $\leq 80.24$ | 32.93 | 100.00 | 6.541 |
| Biased-AO-ARM | | | 1.0 | $\leq 77.94$ | 34.16 | 100.00 | 5.026 |
| LO-ARM-st-st | shared torso | shared torso | 1.0 | $\leq 70.03$ | 90.84 | 100.00 | 4.087 |
| LO-ARM-st-st-topp | shared torso | shared torso | 0.9 | - | 96.02 | 100.00 | 9.237 |
| LO-ARM-st-sep | shared torso | separate | 1.0 | $\leq 68.26$ | 96.26 | **100.00** | **3.229** |
| LO-ARM-st-sep-topp | shared torso | separate | 0.3 | - | 96.70 | 100.00 | 3.859 |

*Table 3.* LO-ARM's perforamance on chemistry-specific metrics on ZINC250k. Specifically, lower **sim**ilarity indicates higher diversity for molecules in the dataset. On all these metrics, LO-ARM exceeds or matches the performance of the state-of-the-art results of the models. The results of other methods are cited from (Mazuz et al., 2023). **V.** is short for Validity.

| Method | QED↑ | SAS↓ | Sim.↓ | V.%↑ |
|---|---|---|---|---|
| Ground truth (test data) | 0.75 | 2.76 | 0.35 | 100.0 |
| JT-VAE | 0.64 | 4.69 | - | **100.00** |
| GCPN | 0.65 | 4.53 | - | 99.00 |
| Taiga | **0.75** | **2.89** | - | 88.00 |
| LO-ARM-st-sep (ours) | **0.75** | 3.08 | **0.34** | 96.26 |

address similar problems. Specifically, the Top-$p$ sampler only draws from dimensions with high probability, and $p$ corresponds to the cumulative probability of the top dimensions ranked with respect to their probabilities. Specifically, $p = 1.0$ regresses to sampling the entire distribution, and the lower the $p$, the greedier the sampler. We tested this approach by sharpening the distribution of the classifier $f_\theta$ after the policy has sampled a dimension. After incorporating the Top-$p$ sampler, we can see that both the validity of Biased-AO-ARM-topp (Top-$p = 0.9$) and LO-ARM-st-st-topp (Top-$p = 0.9$) increase by a large margin, while at a cost of downgraded FCDs. As expected, the Top-$p$ sampling introduces a small bias, shifting the model's distribution away from the true data distribution. We provide further ablation studies on this effect in Appendix D.

Through incorporating Top-$p$ sampling at test time, LO-ARM-st-st-topp achieves similar performance as LO-ARM-st-sep on validity and uniqueness. While LO-ARM-st-sep is a more performant generative model in terms of FCD, LO-ARM-st-st-topp is more computationally efficient for training without using a separate network. As for inference, their cost would be close as the variational order policy is not needed. Both options could be useful in practice, depending on actual needs.

Interestingly, we find that LO-ARM-st-sep is more robust to Top-$p$ sampling. Even if we set $p = 0.3$, the downgrade of its FCD is marginal (from 3.229 to 3.859, staying the SOTA). This suggests that the dimensions sampled with the model order policy have highly concentrated probability mass, which therefore implies that the order policy favours dimensions with higher certainty. This aligns with our observation presented in Figure 1.

Thirdly, on the front of practical usefulness, the ultimate goal of the generative model is is to sample from the same chemical space that the training data comes from. On the other hand, the validity and uniqueness metrics are fairly saturated by existing methods, and the FCD on its own is not a strong enough predictor of molecule quality. To address this issue, we further evaluate LO-ARM-st-sep on the chemistry-specific metrics, including 1) Synthetic Accessibility Score (SAS), which measures the ease of synthesizing a chemical compound (i.e., the lower the score, the easier the compound to be synthesized), 2) Quantitative Estimate of Drug-likeness (QED) which evaluates how well a compound's physicochemical properties align with those of successful administered drugs (the higher the score, the more aligned), and 3) pairwise similarity measures the diversity in samples (lower score corresponds to higher diversity). We visualize the distributions on these three metrics in Figure 2.

On both datasets the distributions of these metrics calculated on the LO-ARM samples closely match the corresponding ground-truth (GT) data, a finding consistent with their favorable FCD scores. We also compare LO-ARM with other methods in the literature on these metrics. As shown in Table 3, LO-ARM exceeds or matches the previous best results in all three metrics.

Finally, we visualize the generation ordering learned by LO-ARM on ZINC250k in Figure 1 and Appendix B.2. Without imposing any inductive bias on orderings, LO-ARM learns to favour an edge-first ordering, i.e, 1) building skeleton of a molecule with chemical bonds, 2) infilling the rest of the adjacency matrix "no-edge" states or the "imaginery" bonds, and finally 3) infilling atoms (See Appendix B.2). This is different from the node-first ordering proposed in Kong et al. (2023). To analyze the consistency of this learned ordering, we sample $30,000$ molecules with three LO-ARM-st-sep models trained with different random seeds, and find that $99.9\%$ of the molecules are generated with the edge-first ordering (See Appendix C). Note that learning a context-dependent ordering (that varies across different data/molecules) within the edge or node dimensions is still critical to performance, as we can see by comparing LO-ARM with Biased-AO-ARM which just uniformly samples within edge or node dimensions.

## 6. Conclusions and Discussion

We have introduced LO-ARM, a novel autoregressive method that can learn the ordering for generating data without canonical ordering. We derive a simple variational lower bound which can be optimized with unbiased stochastic gradients. In addition, we design a scalable training algorithm which makes LO-ARM easily adaptable to real-world tasks. Moreover, we evaluate LO-ARM on two popular molecular generation tasks, and achieve state-of-the-art results in terms of FCD and QED, and competitive results on molecule validity, uniqueness and SAS. In particular, we show that LO-ARM can learn a consistent ordering for generating new molecules without requiring inductive bias on generation orderings, which in turn contributes to a significant performance gain for generative modeling.

On the task of molecule generation specifically, although LO-ARM yields a more performant generative model in terms of FCD, there is still room to improve the validity of individual molecules compared to diffusion-based methods. We hypothesize that these models may have learned to denoise fragment-like sub-graphs. With LO-ARM, one way to do this is to incorporate more complex molecular constraints into the ordering policies. Another improvement could be to use a more memory efficient representation of graphs, such that we do not need to store the full adjacency matrix. On a general note, we are also interested in scaling up LO-ARM to other domains of higher data dimensions, e.g., high-resolution image generation and protein design.

## Impact Statement

This paper presents the work whose goal is to advance the field of Machine Learning. There are many potential societal consequences of this work, none of which we feel must be specifically highlighted here.

## Acknowledgements

We would like to extend deep thanks to Yee Whye Teh, Arnaud Doucet and Yujia Li for diligently reviewing this paper and providing valuable comments.

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

*Figure 2.* Evaluating LO-ARM on (a) ZINC250k and (b) QM9) on the chemistry-specific metrics , i.e, Synthetic Accessibility Score (SAS), Quantitative Estimate of Drug-likeness (QED), and pairwise similarity between samples. On both datasets, the distributions of the samples generated with LO-ARM on these metrics are well-matched with the corresponding ground-truth (GT) data, which is well-aligned with their performance on FCD scores.

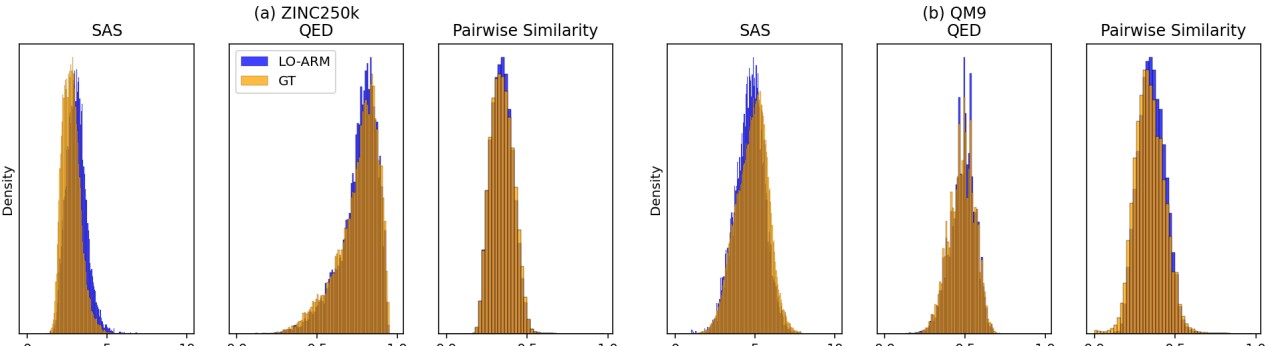

Gu, J., Liu, Q., and Cho, K. Insertion-based decoding with automatically inferred generation order. *Transactions of the Association for Computational Linguistics*, 7:661–676, 11 2019. ISSN 2307-387X. doi: 10.1162/tacl_a_00292. URL https://doi.org/10.1162/tacl_a_00292.

Holtzman, A., Buys, J., Du, L., Forbes, M., and Choi, Y. The curious case of neural text degeneration. In *International Conference on Learning Representations*.

Hoogeboom, E., Gritsenko, A. A., Bastings, J., Poole, B., van den Berg, R., and Salimans, T. Autoregressive diffusion models. In *International Conference on Learning Representations*, 2022.

Jo, J., Lee, S., and Hwang, S. J. Score-based generative modeling of graphs via the system of stochastic differential equations. In Chaudhuri, K., Jegelka, S., Song, L., Szepesvari, C., Niu, G., and Sabato, S. (eds.), *Proceedings of the 39th International Conference on Machine Learning*, volume 162 of *Proceedings of Machine Learning Research*, pp. 10362–10383. PMLR, 17–23 Jul 2022.

Kingma, D. P. and Welling, M. Auto-encoding variational Bayes. In *International Conference on Learning Representations*, 2014.

Kong, L., Cui, J., Sun, H., Zhuang, Y., Prakash, B. A., and Zhang, C. Autoregressive diffusion model for graph generation. In *International Conference on Machine Learning*, pp. 17391–17408. PMLR, 2023.

Kool, W., Hoof, H. V., and Welling, M. Buy 4 REINFORCE samples, get a baseline for free! In *DeepRLStructPred@ICLR*, 2019a.

Kool, W., van Hoof, H., and Welling, M. Stochastic Beams and Where To Find Them: The Gumbel-Top-k Trick for Sampling Sequences Without Replacement. In *Proceedings of the 36th International Conference on Machine Learning*, pp. 3499–3508. PMLR, 2019b.

Langley, P. Crafting papers on machine learning. In Langley, P. (ed.), *Proceedings of the 17th International Conference on Machine Learning (ICML 2000)*, pp. 1207–1216, Stanford, CA, 2000. Morgan Kaufmann.

Li, X., Trabucco, B., Park, D. H., Luo, M., Shen, S., Darrell, T., and Gao, Y. Discovering non-monotonic autoregressive orderings with variational inference. In *International Conference on Learning Representations*, 2021. URL https://openreview.net/forum?id=jP1vTH3inC.

Li, Y., Vinyals, O., Dyer, C., Pascanu, R., and Battaglia, P. Learning deep generative models of graphs. *arXiv preprint arXiv:1803.03324*, 2018.

Lou, A., Meng, C., and Ermon, S. Discrete diffusion modeling by estimating the ratios of the data distribution. *International Conference on Machine Learning*, 2024.

Mazuz, E., Shtar, G., Shapira, B., and Rokach, L. Molecule generation using transformers and policy gradient reinforcement learning. *Scientific Reports*, 13(1):8799, 2023.

Ou, J., Nie, S., Xue, K., Zhu, F., Sun, J., Li, Z., and Li, C. Your absorbing discrete diffusion secretly models the conditional distributions of clean data. *arXiv preprint arXiv:2406.03736*, 2024.

Plackett, R. L. The analysis of permutations. *Journal of the Royal Statistical Society. Series C (Applied Statistics)*, 24 (2):193–202, 1975.

Preuer, K., Renz, P., Unterthiner, T., Hochreiter, S., and Klambauer, G. Fréchet chemnet distance: A metric for

generative models for molecules in drug discovery. *Journal of Chemical Information and Modeling*, 58(9):1736–1741, 2018.

Ronneberger, O., Fischer, P., and Brox, T. U-net: Convolutional networks for biomedical image segmentation. In Navab, N., Hornegger, J., Wells, W. M., and Frangi, A. F. (eds.), *Medical Image Computing and Computer-Assisted Intervention – MICCAI 2015*, pp. 234–241, Cham, 2015. Springer International Publishing.

Sahoo, S. S., Arriola, M., Schiff, Y., Gokaslan, A., Marroquin, E., Chiu, J. T., Rush, A., and Kuleshov, V. Simple and effective masked diffusion language models. *arXiv preprint arXiv:2406.07524*, 2024.

Salimans, T. and Knowles, D. A. On using control variates with stochastic approximation for variational Bayes and its connection to stochastic linear regression. *arXiv preprint arXiv:1401.1022*, 2014.

Shi, J., Zhou, Y., Hwang, J., Titsias, K. M., and Mackey, L. Gradient estimation with discrete Stein operators. In *Advances in Neural Information Processing Systems*, 2022.

Shi, J., Han, K., Wang, Z., Doucet, A., and Titsias, M. K. Simplified and generalized masked diffusion for discrete data. In *Advances in Neural Information Processing Systems*, 2024.

Uria, B., Murray, I., and Larochelle, H. A deep and tractable density estimator. In Xing, E. P. and Jebara, T. (eds.), *Proceedings of the 31st International Conference on Machine Learning*, volume 32 of *Proceedings of Machine Learning Research*, pp. 467–475, Bejing, China, 22–24 Jun 2014. PMLR.

Vignac, C., Krawczuk, I., Siraudin, A., Wang, B., Cevher, V., and Frossard, P. Digress: Discrete denoising diffusion for graph generation. In *International Conference on Learning Representations*, 2023.

Welleck, S., Brantley, K., Iii, H. D., and Cho, K. Non-monotonic sequential text generation. In Chaudhuri, K. and Salakhutdinov, R. (eds.), *Proceedings of the 36th International Conference on Machine Learning*, volume 97 of *Proceedings of Machine Learning Research*, pp. 6716–6726. PMLR, 09–15 Jun 2019. URL https://proceedings.mlr.press/v97/welleck19a.html.

Xu, K., Hu, W., Leskovec, J., and Jegelka, S. How powerful are graph neural networks? In *International Conference on Learning Representations*, 2019.

Yang, Z., Dai, Z., Yang, Y., Carbonell, J., Salakhutdinov, R. R., and Le, Q. V. Xlnet: Generalized autoregressive pretraining for language understanding. In Wallach, H., Larochelle, H., Beygelzimer, A., d'Alché-Buc, F., Fox, E., and Garnett, R. (eds.), *Advances in Neural Information Processing Systems*, volume 32. Curran Associates, Inc., 2019.

You, J., Ying, R., Ren, X., Hamilton, W., and Leskovec, J. Graphrnn: Generating realistic graphs with deep autoregressive models. In *International Conference on Machine Learning*, pp. 5708–5717. PMLR, 2018.

# A. A Toy Example of Generating MNIST with LO-ARM

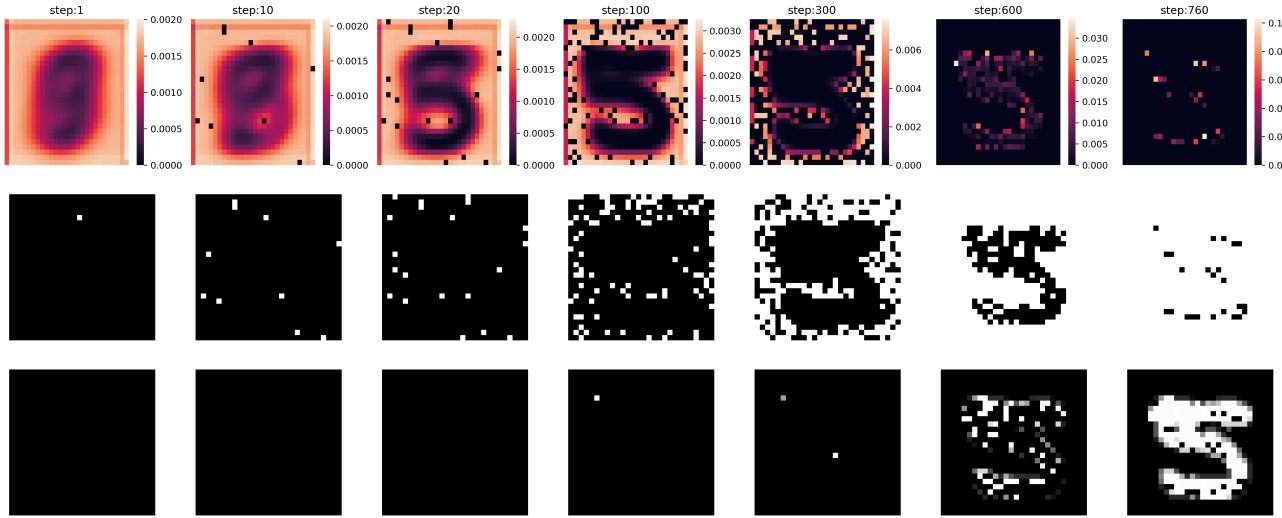

*Figure 3.* An example of data generation in MNIST using LO-ARM with shared torso parametrization of the order-policy distribution. Each column visualizes a step of the sampling process (steps shown are $i = 1, 10, 20, 100, 300, 600, 760$). The panels in the first row show the order-policy values $p_\theta(z_i|\mathbf{z}_{<i}, \mathbf{x}_{\mathbf{z}_{<i}})$ at step $i$ depicted as an $28 \times 28$ image where dark colour indicates small probability value and light colour higher value. The plot in the second row shows all sampled pixel locations (immediately after having determined the current value for $z_i$) depicted as binary masks. Third row shows the progression of generating the actual digit. The figure highlights that the learned order-policy prioritizes unmasking pixels with higher certainty. The boundary-to-center infilling strategy, shown in both the prioritization via the order-policy (first row) and the unmasking sequence (second row), illustrates this. The rough generation order — borders, background, then digit — indicates a clear preference for areas of increasing certainty. Initially, the model confidently fills in the black boundaries and background (third row), leading to the gradual, confident clarification of the digit into "5".

In addition to our main quantitative evaluation with molecular graph data, we also provide an illustrative example of generating a new MNIST sample (Figure 3). Firstly, to cast image generation as generative modelling with discrete tokens, we treat the 256 gray-scale values as discrete categories and augment the token space with one additional mask token. Then generating a new MNIST sample autogressively (one token at a time) would be similar to generating with masked diffusion models ((Austin et al., 2021; Hoogeboom et al., 2022; Shi et al., 2024; Sahoo et al., 2024)), and the main difference is LO-ARM only generates one token at a time, while masked diffusion models could generate multiple tokens at one denoising step. As a result, the total number of sampling steps equals to the total number of pixels in one MINST image.

Next, we train the model with Algorithm 1. Specifically, we parameterize the categorical classifier with a UNet (Ronneberger et al., 2015), parametrize the both the variational and order-policy with shared-torso (see Section 3.3 and Appendix F).

Finally, we illustrate the learning order ARM (LO-ARM) algorithm in MNIST and we visualize probabilistic orders when sampling from the model, using the optimized model we apply Algorithm 2 to generate new digits, where we inspect also the autoregressive order the pixels are generated. Figure 3 illustrates different steps of the sampling process; see figure caption for details. We can observe that LO-ARM, which parametrizes the posterior order-policy with shared-torso, prefers to sample first the border pixel values and then the "digit" pixels in the centered foreground. Presumably, the reason is that the model learns (through ELBO maximization) to prefer an order where "easier" border pixel locations tend to be specified first. Note that there is no inductive bias towards orders that prioritise first more confident data dimensions.

# B. Gallery of Generated Molecules

### B.1. Individual Molecules

We show the QM9 (Figure 4) and ZINC250K (Figure 5) samples generated with our best LO-ARM models presented in Table 1 and 2.

*Figure 4.* Generated QM9 samples.

*Figure 5.* Generated ZINC250K samples.

### B.2. How LO-ARM Generates A Molecule

We provide the full generation path for the molecule presented in Figure 1. In particular, we only show the full path of building skeleton (Stage 1 in Figure 6) and infilling atoms (Stage 3 in Figure 7), as Stage 2 only contains repetitive visible molecules.

## C. Consistency Analysis of Learned Ordering for Molecular Graph Generation

As we can see in the case study in Figure 1 and Appendix B.1, LO-ARM prefers a simple and specific ordering to generate new samples, which unfolds into three stages: 1) building the skeleton of molecules, 2) infilling the "imaginery" bonds in the adjacency matrix, and 3) infilling atoms. With this observation, we can actually simplify analyzing its consistency to a problem of template matching. Specifically, the template matching unfolds into the following steps:

- Firstly, when generating a sample, we label the token generated at each step as one of the three states, i.e., N for node/atom, E for edge/bond and N for no_edge or imaginary bond, and output the token state sequences.

- Secondly, for each state sequence, we compress it to only keep the transitions between different states. For example, a state sequence EEEANNNAAA will be compressed to EANA, and the expected ordering sequence is compressed to ENA.

- Finally, we count the exact matchings between the compressed state sequences and the template of expected ordering sequence, i.e., ENA.

To test the consistency statistically, firstly, we run the experiments to train LO-ARM with three different seeds. Secondly, with each of the three models, we generate 10000 molecules and store their corresponding token state sequences. Note that,

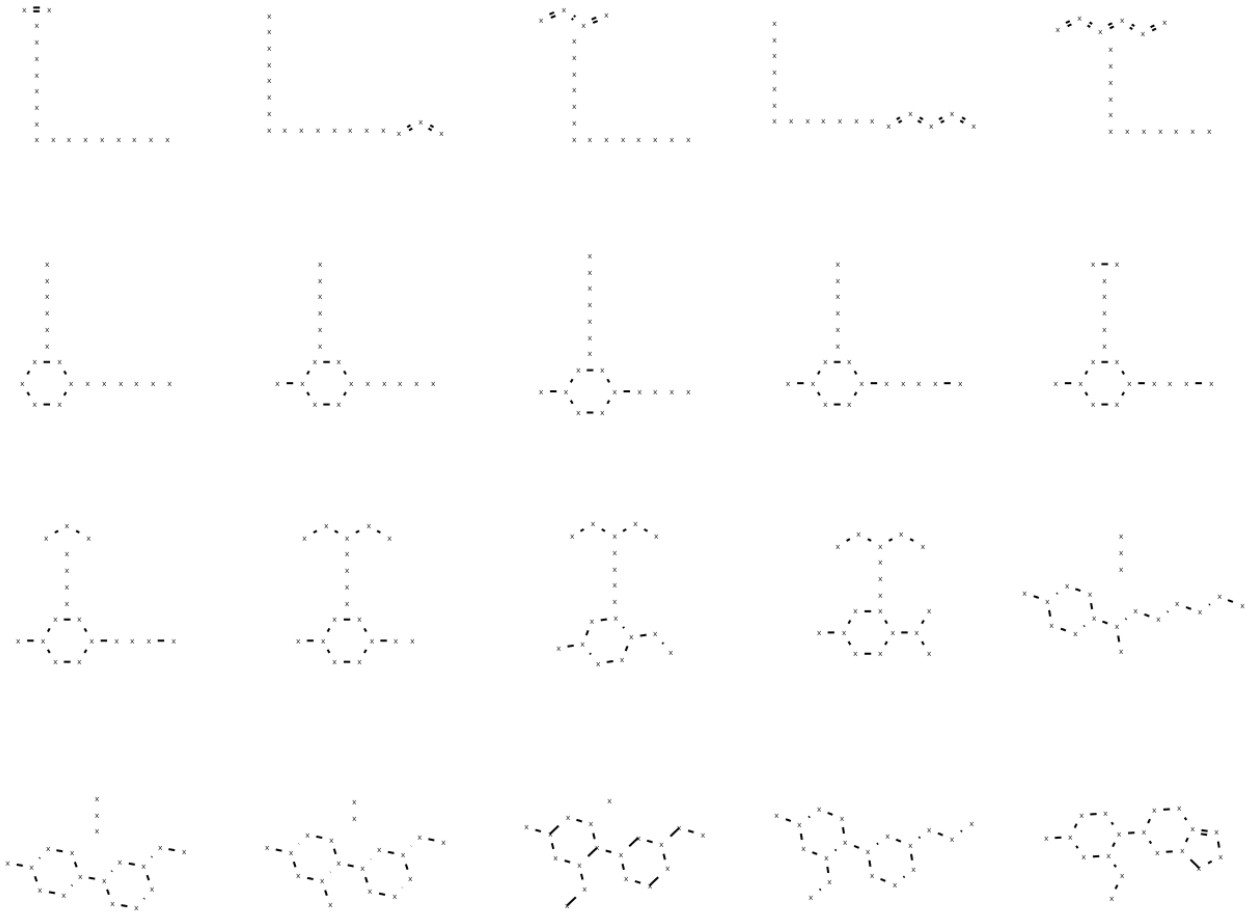

*Figure 6.* The full sample path of building the skeleton of the molecule presented in Figure 1. Masked nodes/atoms are labelled with x, and the bonds in the adjacency are unmasked one by one.

we do not filter invalid molecules in this analysis. Finally, for these 30000 state sequences, we count the matchings with the expected ordering through the algorithm described above. We find that $99.9\%$ (29973 out of 30000) of the orderings match the ordering pattern that we have observed.

## D. Detailed Results

We provide a detailed ablation analysis for the architecture options described in Section 3.3 and Appendix F for both QM9 Table 4 and Table 5. In addition, for the experiments with ZINC250k we also provide the results of varying the threshold $p$ in the Top-$p$ (Shi et al., 2024) sampler aforementioned in Section 5.

## E. Experiment Details for Molecular Graph Generation

We provide the details of our experimentation with the QM9 and ZINC250K datasets. The statistics of these two datasets are summarized in Table 6.

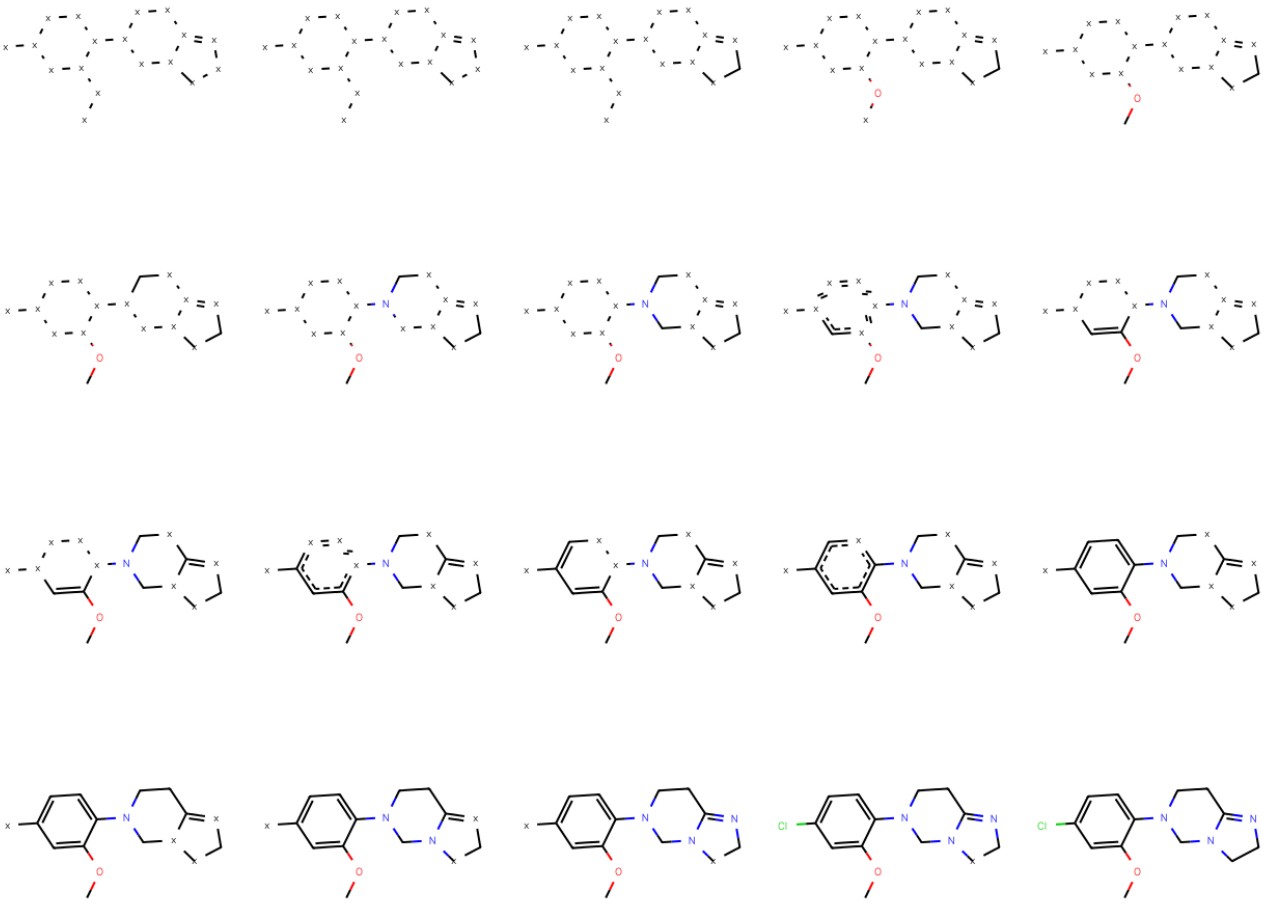

*Figure 7.* The full sample path of infilling atoms of the molecule presented in Figure 1. We also include the last two steps of Stage 2, i.e., infilling the no_edge states at the beginning of this figure. During the generation process, atoms are unmasked one by one.

### E.1. Data Representation

We follow the standard setup (Vignac et al., 2023; Eijkelboom et al., 2024) to transform raw data represented as SMILES (Simplified Molecular Input Line Entry System) strings to dense graphs, $\mathbf{G} = (\mathbf{H}_n, \mathbf{H}_e, \mathbf{H}_m)$, where $\mathbf{H}_n$ is the node vector, $\mathbf{H}_e$ is the edge adjacency matrix and $\mathbf{H}_m$ is the node mask. In particular, the node masks pad molecules with variable sizes to a static size, such that they can be processed with mini-batches. Moreover, same as DiGress, to represent the edge connections with a dense adjacency matrix, an "imaginery" bond is introduced, which we also call no_edge. This imaginary bond has no chemical effect when building molecules but helps us distinguish from the entries masked different masks. Now we introduce the *masks* used in our implementation.

As discussed in Section 2, we model autoregressive generation as unmasking process. To do this, we augment the token vocabulary with one special token to represent the mask, which we call *sampling mask*. Specifically, for molecular graph generation, we augment vocabularies of both nodes and edges with one zero-valued token respectively. Note that, although both these tokens are zeros in inputs, they may have different latent embeddings as the graph transformer processes nodes and edges differently, as introduced in the next section. Notation-wise, in this paper, we call the node mask $\mathbf{H}_m$ attention mask to distinguish from the sampling mask.

*Table 4.* Detailed results on on QM9. The negative loglikelihoods (NLLs) of our methods are evaluated against the test set, and other metrics are calculated with the generated samples with the corresponding methods.

| Method | $p_\theta$ | $q_\theta$ | NLL↓ | Validity%↑ | Uniqueness%↑ | FCD↓ |
|---|---|---|---|---|---|---|
| Uniform-ARM | | | $\leq 24.56 \pm 0.12$ | $98.81 \pm 0.09$ | $99.08 \pm 0.03$ | $0.691 \pm 0.018$ |
| LO-ARM-ent-st | entropy | shared torso | $\leq 24.05 \pm 0.29$ | $99.04 \pm 0.15$ | $99.13 \pm 0.17$ | $0.645 \pm 0.036$ |
| LO-ARM-ent-sep | entropy | separate | $\leq 23.70 \pm 0.11$ | $98.93 \pm 0.14$ | $99.03 \pm 0.15$ | $0.444 \pm 0.013$ |
| LO-ARM-st-st | shared torso | shared torso | $\leq 22.23 \pm 0.01$ | $99.03 \pm 0.08$ | $98.49 \pm 0.17$ | $0.465 \pm 0.027$ |
| LO-ARM-st-sep | shared torso | separate | $\leq 21.22 \pm 0.47$ | $99.77 \pm 0.08$ | $98.71 \pm 0.10$ | $0.264 \pm 0.031$ |

*Table 5.* Molecule generation on ZINC250K. In particular, the NLLs of the LO-ARMs with the Top-$p$ sampling are omitted as the sampler is only used in inference time and does not change the training-time metrics.

| Method | Top-$p$ | NLL↓ | Validity%↑ | Uniqueness%↑ | FCD↓ |
|---|---|---|---|---|---|
| Uniform-ARM | 1.0 | $\leq 80.08 \pm 0.44$ | $31.40 \pm 1.19$ | $100.00 \pm 0.00$ | $6.587 \pm 0.070$ |
| Biased-Uniform-ARM | 1.0 | $\leq 78.16 \pm 0.27$ | $33.68 \pm 0.35$ | $100.00 \pm 0.00$ | $5.075 \pm 0.051$ |
| Biased-Uniform-ARM-topp | 0.9 | - | $79.41 \pm 1.03$ | $100.00 \pm 0.00$ | $22.492 \pm 0.227$ |
| LO-ARM-ent-st | 1.0 | $\leq 77.92 \pm 0.11$ | $41.36 \pm 0.42$ | $100.00 \pm 0.00$ | $4.776 \pm 0.090$ |
| LO-ARM-ent-sep | 1.0 | $\leq 77.04 \pm 0.22$ | $41.50 \pm 0.99$ | $100.00 \pm 0.00$ | $4.770 \pm 0.104$ |
| LO-ARM-st-st | 1.0 | $\leq 69.92 \pm 0.67$ | $90.58 \pm 0.35$ | $100.00 \pm 0.00$ | $4.122 \pm 0.039$ |
| LO-ARM-st-st-topp | 0.9 | - | $96.36 \pm 0.45$ | $100.00 \pm 0.00$ | $9.388 \pm 0.100$ |
| LO-ARM-st-st-topp | 0.75 | - | $97.44 \pm 0.45$ | $99.99 \pm 0.02$ | $16.416 \pm 0.070$ |
| LO-ARM-st-sep | 1.0 | $\leq 68.91 \pm 0.45$ | $95.97 \pm 0.27$ | $99.99 \pm 0.01$ | $3.33 \pm 0.130$ |
| LO-ARM-st-sep-topp | 0.9 | - | $96.16 \pm 0.48$ | $100.00 \pm 0.00$ | $3.586 \pm 0.118$ |
| LO-ARM-st-sep-topp | 0.3 | - | $96.56 \pm 0.13$ | $100.00 \pm 0.00$ | $3.852 \pm 0.035$ |

## E.2. Parameterizing Classifier, Posterior and Variational Order Policies

To ensure a fair comparison with the baselines, we employ the same graph transformer network which is used in (Vignac et al., 2023; Eijkelboom et al., 2024). Just as done in DiGress and CatFlow, our graph transformer takes as a *masked* graph $\bar{\mathbf{G}} = (\bar{\mathbf{H}}_n, \bar{\mathbf{H}}_e, \mathbf{H}_m)$ and predicts a distribution over the clean graphs, represented as $\hat{\mathbf{G}} = (\hat{\mathbf{H}}_n, \hat{\mathbf{H}}_e)$. Again, the attention mask $\mathbf{H}_m$ stays constant during whole computation. For our classifier, we implement the shared-torso parameterization for the posterior order policy $p_\theta(\cdot|\boldsymbol{x}_{\boldsymbol{z}_{<i}})$ through augmenting the output layers of node and edge logits with one extra dimension respectively, the output of which are used the logits of the posterior order policy, and each logit corresponds to the position of a node or edge. Then during sampling, we obtain the logits of the posterior order policy through concatenating the extra node logits and the flattened extra edge logits.

For the variational order-policy , its shared-torso parameterization is the same as the posterior order-policy . Moreover, when implementing it with a separate neural network, we employ a smaller graph transformer with the same architecture. This is because the variational order-policy also takes graphs as inputs, i.e., $q_\theta(\cdot|\boldsymbol{x} = \mathbf{G})$, and we want to leverage the nice equivariance properties (Vignac et al., 2023) brought by the graph transformer to improve sample efficiency. Moreover, intuitively, in addition to homogeneous inputs, the architectural homogeneity would also help optimize the KL term between the variational and posterior order policies. Finally, the output dimension the $q_\theta$ network is only one for both node and edge logits. Same as the $p_\theta$ network, we obtain the output logits after flattening and concatenating them. Moreover, to enforce symmetry of the adjacency matrix,

Note that, there may just be our design considerations specific to the task molecular graph generation. In principle, we pose no constraints on designing the three components. There could be more efficient parameterization choices, and we will explore as part of the follow-up work.

## E.3. Enforcing Symmetry in Masking and Unmasking

Recall that during training 1, after sampling an latent ordering $\boldsymbol{z} = (\boldsymbol{z}_{<i}, \boldsymbol{z}_{\geq i})$, where $\boldsymbol{z}_{<i}$ is a partial ordering, and $\boldsymbol{z}_{\geq i}$ is the unordered set, and we need to mask the tokens at dimensions $\boldsymbol{z}_{\geq i}$. To enforce symmetry of the adjacency matrix $\hat{\mathbf{H}}_e$, for edges, we only process the dimensions in the order-policy corresponding to the upper half of $\mathbf{H}_m$, with the lower half being masked out according to the attention mask. After zeroing out $\boldsymbol{z}_{\geq i}$, we flip the upper half to the lower half. We

*Table 6.* Statistics of QM9 and ZINK250k datasets used in our molecule generation tasks.

| Dataset | Number of molecules | Number of nodes | Number of node types | Number of edge types |
|---------|---------------------|-----------------|----------------------|----------------------|
| QM9 | $133,885$ | $1 \leq |\mathcal{V}| \leq 9$ | 4 | 4 |
| ZINC250k | $249,455$ | $6 \leq |\mathcal{V}| \leq 38$ | 9 | 4 |

*Figure 8.* Illustration of the process of masking molecular graphs. A graph is represented as a node vector and an dense adjacency matrix which is symmetric with respect to the diagonal. The upper row indicates the active dimensions (green blocks) in the order-policy , and the lower row shows the corresponding masked graph. When an edge dimension is sampled (yellow block), to ensure symmetry, we mask both the corresponding data dimensions (blue blocks) in the graph and its symmetric dimension. Note that sampling an ordering from the order-policy is a sampling-without-replacement problem, and once a dimension is sampled in the order-policy , we mask it out with the sampling mask (red blocks).

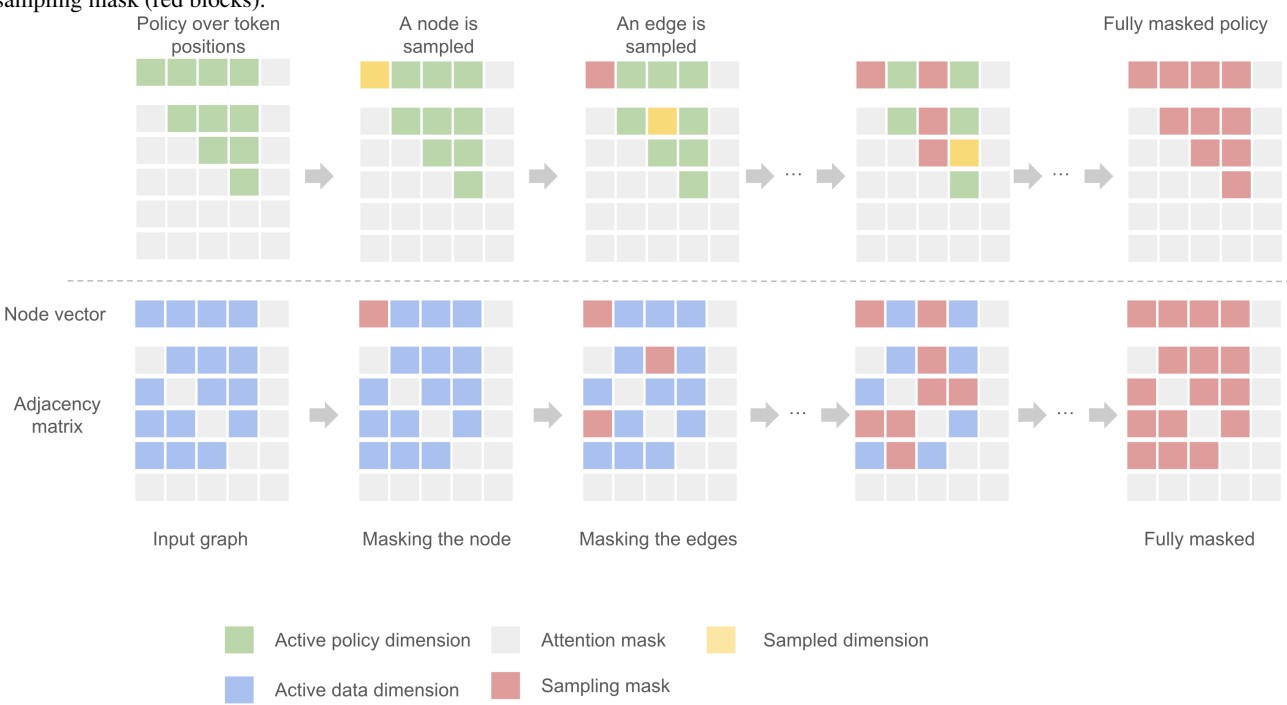

always perform this operation regardless whether an edge is sampled or not. Note that, except for the transformer outputting nodes and edges separately, we provide no inductive bias about edge and node positions to the order-policy . We provide an illustration of the masking process in Figure 8.

We apply similar process during sampling Algorithm 2 as shown in Figure 9. In particular, instead of starting with a fully masked graph with a full adjacency matrix, we only consider the upper half of the adjacency matrix, and flipping the upper half after one sampling step. In this way, we will always obtain a symmetric adjacency matrix regardless whether an edge is unmasked.

### E.4. Experimental Setup

We report the hyperparameters in Table 7. Moreover, the hidden dimensions for the classifier network are kept the same as (Vignac et al., 2023) and all data is kept the same as in (Jo et al., 2022). All experiments were run until convergence.

## F. Options of Model Parameterization and further details about REINFORCE LOO

We provide the details of different choices of parameterizing the distributions needed in Algorithm 1 and Algorithm 2.

As mentioned in the main paper, we assume that data are discrete and each dimension (*token*) $x_i$ takes $m$ categorical values.

*Figure 9.* Illustration of the generation process as unmasking for molecular graphs. Generating a new molecular graph reserves the masking process. The only difference is that for the molecule graph, we start with the upper half of the adjacency matrix, and each time an edge dimension is sampled with the order-policy , we unmask the corresponding data dimension in the adjacency matrix and its symmetric dimension.

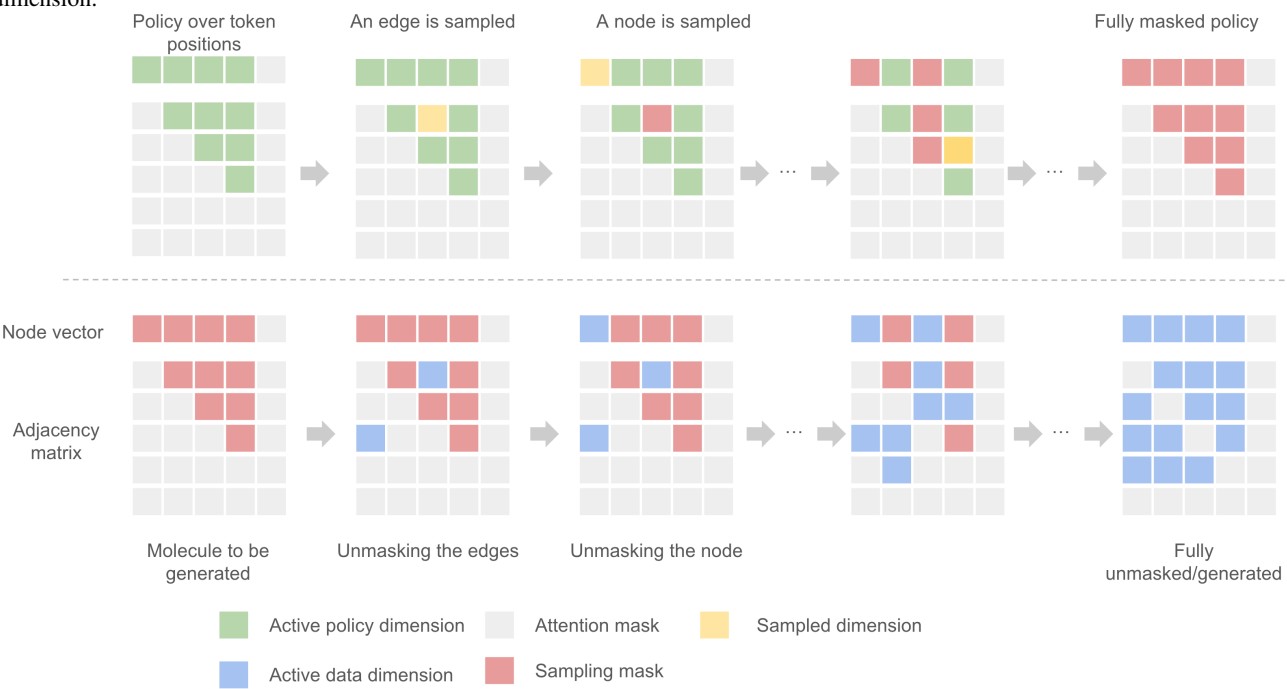

*Table 7.* Hyperparameter setup.

| Hyperparameter | QM9 | ZINC250k |
|---|---|---|
| Optimizer | AdamW | AdamW |
| Scheduler | Cosine Annealing | Cosine Annealing |
| Learning Rate | $1 \cdot 10^{-5}$ | $1.5 \cdot 10^{-5}$ |
| Weight Decay | $1 \cdot 10^{12}$ | $1 \cdot 10^{12}$ |
| EMA | 0.9999 | 0.9999 |

We represent each $x_i$ as an $m+1$-dimensional (rather than $m$-dimensional) one-hot vector where the final $m+1$-th value is a special one since it indicates that the category of $x_i$ is currently *masked* or unspecified.

We assume a Neural Network (NN) that takes $L$ inputs, i.e., $L$ $m+1$-dimensional one-hot vectors of the discrete tokens, and gives in the output $L$ vectors of classifier logits, each corresponding to the $m$ categorical values (excluding the $m+1$-th masked value) for each token $x_i$. More precisely, we denote the NN output logit vectors by

$$f_{\theta, z_i = k}(\bar{\boldsymbol{x}}_{\boldsymbol{z}_{<i}}) \in \mathbb{R}^m, \ k = 1, \ldots, L,$$

so that each $f_{\theta, z_i = k}$ consists of the logits of the $z_i$-th output classifier for $z_i = k$. The NN takes as input a $L \times (m+1)$ matrix $\bar{\boldsymbol{x}}_{\boldsymbol{z}_{<i}}$ which in all indices $\boldsymbol{z}_{<i}$ contains the corresponding observed (or generated) data values $\boldsymbol{x}_{\boldsymbol{z}_{<i}}$, that precede token $z_i$, and in the remaining indices $\boldsymbol{z}_{\geq i}$ is filled in with the mask token value, i.e., with the one-hot vector that has the value 1 in the final $m+1$-th dimension. Another way to view this is that $\bar{\boldsymbol{x}}_{\boldsymbol{z}_{<i}}$ is a modification of $\boldsymbol{x}$ that has all $\boldsymbol{z}_{<i}$ tokens of $\boldsymbol{x}$ unmasked and the remaining $\boldsymbol{z}_{\geq i}$ tokens of $\boldsymbol{x}$ masked. Then, based on these NN outputs the probability distribution that specifies the value for $x_{z_i}$ is a categorical distribution or *classifier* of the form

$$p_\theta(x_{z_i} | \boldsymbol{x}_{\boldsymbol{z}_{<i}}) = \mathrm{softmax}(f_{\theta, z_i}(\bar{\boldsymbol{x}}_{\boldsymbol{z}_{<i}})),$$

where the logits are passed through softmax to provide a probability vector. The other quantities needed to be specified are the model order-policy factors $p_\theta(z_i = k | \boldsymbol{z}_{<i}, \boldsymbol{x}_{\boldsymbol{z}_{<i}})$ and the corresponding variational *posterior-order* distribution factor

$q_\theta(z_i = k|\mathbf{z}_{<i}, \mathbf{x})$. As mentioned in the main text, the difference between these two distributions is that the first can exploit information only from the previous specified dimensions $\mathbf{x}_{\mathbf{z}_{<i}}$, while the second can condition on the full $\mathbf{x}$. We specify these distributions as described below.

### F.1. Form of Order-Policy Distribution

We explore two choices for parametrizing each order-policy factor $p_\theta(z_i|\mathbf{z}_{<i}, \mathbf{x}_{\mathbf{z}_{<i}})$. In the first option we re-use the NN classifiers $p_\theta(x_{z_i}|\mathbf{x}_{\mathbf{z}_{<i}})$, as defined above, and we form $p_\theta(z_i = k|\mathbf{z}_{<i}, \mathbf{x}_{\mathbf{z}_{<i}})$ using entropy-based uncertainties. In the second option we model $p_\theta(z_i|\mathbf{z}_{<i}, \mathbf{x}_{\mathbf{z}_{<i}})$ by adding an extra final linear layer to the NN feature representation. This latter approach requires no modification or extension of the NN architecture. We detail both cases below.

**Entropy-based parametrization.** In the first option we incorporate some inductive bias towards more confident or less confident tokens under the model. More precisely, we use entropy-based uncertainty logits so that each factor is written as

$$p_\theta(z_i = k|\mathbf{z}_{<i}, \mathbf{x}_{\mathbf{z}_{<i}}) = \frac{e^{-\beta\mathcal{H}(p_\theta(x_k|\mathbf{x}_{\mathbf{z}_{<i}}))}}{\sum_{k' \in \mathbf{z}_{\geq i}} e^{-\beta\mathcal{H}(p_\theta(x_{k'}|\mathbf{x}_{\mathbf{z}_{<i}}))}}$$

$$= \frac{e^{-\beta\mathcal{H}(\mathrm{softmax}(f_{\theta,k}(\bar{\mathbf{x}}_{\mathbf{z}_{<i}, \mathbf{z}_{\geq i}})))}}{\sum_{k' \in \mathbf{z}_{\geq i}} e^{-\beta\mathcal{H}(\mathrm{softmax}(f_{\theta,k'}(\bar{\mathbf{x}}_{\mathbf{z}_{<i}, \mathbf{z}_{\geq i}})))}}$$

where $\mathcal{H}$ is the entropy of a distribution and $\beta \in \mathbb{R}$ is a scalar parameter. For $\beta = 0$ this becomes the uniform distribution over the $L - i + 1$ values in $\mathbf{z}_{\geq i}$. For $\beta > 0$ the distribution favors the selection of $z_i$ values for which the model data conditionals $p_\theta(x_{z_i}|\mathbf{x}_{\mathbf{z}_{<i}})$ are more certain about the actual categorical value $x_{z_i}$ should assign, while similarly for $\beta < 0$ the preference is reversed towards more uncertain tokens. $\beta$ is treated as an additional model parameter that is optimized jointly with $\theta$ using the variational inference method described in Section 3.2.

**Shared-torso parametrization.** For second option of the order-policy, referred to as *shared-torso*, we add an extra final linear layer to the NN with $L$ scalars $h_{\theta,k}(\cdot) \in \mathbb{R}$, $k = 1, \ldots, L$. Then, each order-policy factor is obtained by

$$p_\theta(z_i = k|\mathbf{z}_{<i}, \mathbf{x}_{\mathbf{z}_{<i}}) = \frac{e^{h_{\theta,k}(\bar{\mathbf{x}}_{\mathbf{z}_{<i}})}}{\sum_{k' \in \mathbf{z}_{\geq i}} e^{h_{\theta,k'}(\bar{\mathbf{x}}_{\mathbf{z}_{<i}})}},$$

where the extra model parameters needed to define these $L$ scalar outputs are optimized jointly with the remaining parameters of the NN architecture.

### F.2. Variational Distribution

As explained in the main text, we model each variational factor $q_\theta(z_i = k|\mathbf{z}_{<i}, \mathbf{x}_{\mathbf{z}_{<i}})$ as

$$q_\theta(z_i = k|\mathbf{z}_{<i}, \mathbf{x}) = \frac{e^{g_{\theta,k}(\mathbf{x})}}{\sum_{k' \in \mathbf{z}_{\geq i}} e^{g_{\theta,k'}(\mathbf{x})}} \tag{15}$$

so that we only need to parametrize the vector function

$$g_\theta(\mathbf{x}) \in \mathbb{R}^L.$$

The first option we consider is adding an extra final layer with $L$ outputs as another head to the main neural architecture (this is the shared-torso option). In the second option, and somehow more flexible, we construct a separate NN where in the final layer it outputs $L$ real values to model the vector $g_\theta(\mathbf{x})$.

Computing and sampling from the variational distribution is very fast since it requires a single forward pass with input $\mathbf{x}$ from the NN (either the shared architecture or the separate network) to obtain and store the vector of logits $g_\theta(\mathbf{x})$. Then sampling a full path $\mathbf{z}$ can be done at once in parallel using the Gumbel-top-k trick (Kool et al., 2019b). Note that sampling two paths $\mathbf{z}_{<i}^1, \mathbf{z}_{<i}^2$, from the variational distribution is needed in order to compute the objective in Equation (16); see next.

### F.3. Objective to use with Automatic Differentiation

As discussed in the main text, training is done using an unbiased REINFORCE leave-one-out (RLOO) gradient computed separately for each data point in the minibatch (then all these gradients are averaged over the minibatch). For a single data point this unbiased gradient is given by Equation (11) in Section 3.2.

For implementation convenience there is a way to obtain the unbiased gradient by programming a certain objective function and then apply automatic differentiation to it. Specifically, what is required is to implement the following objective (to be maximized):

$$\frac{L}{2} \left\{ \left( \log q_\theta(\boldsymbol{z}_{<i}^1 | \boldsymbol{x}) - \log q_\theta(\boldsymbol{z}_{<i}^2 | \boldsymbol{x}) \right) stopgrad[\Delta F] + F_\theta(\boldsymbol{z}_{<i}^1, \boldsymbol{x}) + F_\theta(\boldsymbol{z}_{<i}^2, \boldsymbol{x}) \right\}, \tag{16}$$

where $stopgrad[\Delta F]$ stops the gradient computation in the $\Delta F$. However, note that this objective is just a trick to easily obtain the gradient, while the actual objective that we maximize is the ELBO. This means that monitoring convergence is done by computing the following stochastic ELBO (per data point):

$$\mathcal{L}(\theta) = \frac{L}{2} \left( F_\theta(\boldsymbol{z}_{<i}^1, \boldsymbol{x}) + F_\theta(\boldsymbol{z}_{<i}^2, \boldsymbol{x}) \right), \tag{17}$$

which is just the two-sample version of the one-sample stochastic ELBO from Equation (9). Given a training minibatch these stochastic ELBOs are further averaged over the minibatch.

