# OpenReview forum: "Learning-Order Autoregressive Models with Application to Molecular Graph Generation"
_ICML.cc/2025/Conference — ICML 2025 poster_

### Official Review · Reviewer_u8KG · 2025-03-11

**Overall Recommendation:** 3

**Summary:**

This paper introduces a method for learning the order in which discrete elements from a masked set should be generated. The authors assume the generation process begins with a set of masked elements and, in addition to predicted the unmasked element, the model learns a policy which dictates the order in which elements should be unmasked from the set. The methods uses a variational approach and show how the previously proposed REINFORCE leave-one-out estimator can be used to backprop gradients to the variational distribution network. The authors focus the application of their method on molecule generation using two commonly used datasets and also show how the method can be applied to generate MNIST images.

# Update after rebuttal

I thank the authors for answering my questions and providing additional experimental results. I remain unconvinced that molecule generation is the right application for this method, and I think the additional sampling steps required vs. SMILES models remains a problem. However, I think the method is interesting, well-supported and could be useful for other applications, so I will increase my score.

**Claims And Evidence:**

The authors are well supported by their evaluation results, however, please see below for shortcomings with the evaluation.

**Essential References Not Discussed:**

- DeFoG [1] and GruM [2] have both recently been proposed as graph generative models with strong performance on molecule generation using flow-matching and diffusion, respectively.

[1] https://arxiv.org/abs/2410.04263

[2] https://arxiv.org/abs/2302.03596

**Experimental Designs Or Analyses:**

The experiments are well designed and follow previous work in the area.

**Methods And Evaluation Criteria:**

- For molecule generation tasks the authors should also report the diversity (eg. often defined using tanimoto similarity of the fingerprints) and novelty of the generated molecules. This is to help ensure that the model is not sampling only from a restricted part of chemical space.

- Since the validity and uniqueness metrics are fairly saturated by existing methods, and the FCD on its own is not a strong enough predictor of molecule quality, I think it would be important to report metrics which attempt to measure distances to the data distribution. For example comparing the QED, synthetic accessibility, number of aromatic rings, etc. from a set of generated molecules to the training distribution. Ultimately the goal of the generative model is to sample from the same chemical space that the training data comes from.

- In order to make the evaluation more practically meaningful the authors should report baseline results for a SMILES-based transformer model (eg. Molformer, Chemformer or Molecular Transformer would all be simple starting points), using both canonical and SMILES augmented orderings. This type of model is very commonly used in practice and is a very important baseline for autoregressive molecule generation.

**Other Comments Or Suggestions:**

- Should the equation for $q_{theta}$ at the start of page 6 have a $\mathbf{x}$ instead of $\mathbf{x_{z_{\lt i}}}$?

**Other Strengths And Weaknesses:**

- Is each component of graph (ie. nodes and edges) sampled one-at-a-time? Meaning for a graph with $n$ nodes the method requires $n^2 + n$ sampling steps? This seems like a major limitation in comparison to existing methods for autoregressive molecule generation where the length of the sequence is usually only slightly larger than $n$. It would also become a significant bottleneck for scaling the LO-ARM method to larger graphs.

- I think the overall method for learning the order in which items are generated is an interesting idea and a potentially useful framework for other applications, however, I am not convinced that molecule generation is the right application for this. A number of very strong baselines exist for molecule generation and would likely have significantly faster sampling times than the proposed method. Additionally, once a starting point has been defined, the generation order of molecules could be constrained relatively easily by only unmasking atoms which are connected to already unmasked atoms, significantly reducing the space of permutations and making learning the ordering less important.

**Questions For Authors:**

- I am not really clear on how you achieve single step generation for $\mathbf{z}$. Is each component of $\mathbf{z}$ sampled autoregressively, or are all sampled at the same time?

**Relation To Broader Scientific Literature:**

The paper introduces a novel method for generating graph-structured data (or any other data without a specific ordering). The method is related to masked discrete diffusion and flow-matching methods but introduces a variational framework and a different training strategy. The method produces strong results on graph generation and the authors show example generation traces and provide some intuition for the decisions made by the model. However, I think some important baselines have been missed - see above for evaluation and below for additional diffusion and FM graph generation models.

**Theoretical Claims:**

The theoretical claims of the paper are well supported and the method derivation is very clearly presented.

---

> ### Author Rebuttal · Authors · 2025-04-01
>
> We extend our sincere gratitude to the reviewer for their expert insights and valuable suggestions regarding the incorporation of chemistry-specific metrics to enhance our evaluation. We address their points below.
>
> ## 1. Report results against chemistry-specific metrics
> We evaluated our best-performing models, trained on the ZINC250k dataset, using the Synthetic Accessibility Score (SAS), Quantitative Estimation of Drug-likeness (QED), diversity and novelty metrics. Diversity was measured by calculating the pairwise Tanimoto similarity within a set of generated molecules. The table below summarizes the results, including baselines and measurements on ground truth data, and more detailed visual comparisons of the metric distributions between our models and the ground truth data are available at [1]. As the results indicate, the distribution of these metrics in the samples generated by LO-ARM closely resembles that of the ground truth data.
>
> | mean | QED           | SAS           | Diversity     | Novelty     |
> |----------|----------|---------------|---------|--------|
> | Ground truth  | 0.75  | 2.76  | 0.35  | -           |
> | JT-VAE | **0.76** | 3.37 | - | 100.0 |
> | GCPN | 0.61 | 4.62 | - | 100.0
> |MolecularRNN| 0.68         | 3.59        | - | 100.0
> | LO-ARM        | **0.75**  | **3.08**  | 0.34   | 100.0 |
>
> ## 2. Add baselines of SMILES-based transformers
> Consistent with our baselines (e.g., DiGress and CatFlow), we focus on graph representation of molecules in this work. For the reviewer’s interest, we included a SMILES-based baseline in our GuacaMol results (please see our reply to Reviewer Pa4s). Morever, LO-ARM is agnostic to data representations and could also be applied to SMILES strings. As suggested by the reviewer, SMILES would be more practically useful, adapting LO-ARM to SMILES generation is a promising direction for our future work.
>
> ## 3. Time complexity of inference
> Regarding the cost of sampling, we first note that we can skip the edge sampling stage if the order policy determines it as a no-edge dimension. Second, the learned ordering within LO-ARM effectively separates the generation of non-existent bonds (no-edges) from atoms and real bonds. This separation allows us to generate all no-edges in a single inference step without compromising the chemical validity of the generated molecules, substantially reducing inference overhead. We present a comparison of generic and ordering-informed sampling approaches evaluated on the ZINC250k dataset below.
>
> |                   | Validity | Uniquness | FCD   | Avg. sampling steps |
> |-------------------|----------|-----------|-------|---------------------|
> | Generic sampling  | 96.26    | 100.      | 3.229 | 330.7               |
> | Ordering-informed | 96.13    | 100.      | 3.319 | 48.8                |
>
> We plan to further explore this learned ordering to scale up inference in future work. A possible yet more principled direction could be leverating the equivalence between random order ARMs and masked diffusion models to enable parallel generation. Our preliminary experiment shows that we can obtain 90.0% validity in half of the number of steps using the masked diffusion formulation of our model.
>
> ## 4. Suitability of learning ordering for molecular graph generation
> While generation order constraints reduce the permutation space, our results show a clear benefit to learning a data-dependent order. For example, LO-ARM significantly outperforms GraphARM [2], which unmasks a node and its edges in one step, across all metrics.
>
> To ensure a fair comparison with our baseline models, we have only evaluated unconditional generation. To further tailor LO-ARM for molecular graph generation, one could integrate inductive biases into the backward sampling process during training, which we also recognize as a promising avenue for future development of LO-ARM.
>
> ## 5. Should the equation for qtheta at the start of page 6 have a $x$ instead of $x_{z<i}$?
> Yes, this is a typo, thanks for examining our work thoroughly.
>
> ## 6. Is each component of z sampled autoregressively, or are all sampled at the same time?
> To clarify: z is sampled autoregressively via the order-policy for generation, but for the variational distribution during training, all its components are generated in one pass by the q-network using the Gumbel top-k trick [3].
>
> ## 7. Enrich essential references
> We will update the Related Work section with the references you’ve suggested in the final version of the paper.
>
> We hope that our responses have addressed your concerns and questions. If so, we would be grateful if you would reconsider your decision in light of our clarifications and update your recommendation score accordingly. We remain available and eager to address any further concerns you may have.
>
> [1] Chemistry-specific metrics: https://drive.google.com/file/d/1miSlh2vYYRfqHJ-e5QmYFmht5xMQcqEO/view?usp=sharing
>
> [2] https://arxiv.org/pdf/2307.08849
>
> [3] https://arxiv.org/pdf/1903.06059

---

### Official Review · Reviewer_Pa4s · 2025-03-12

**Overall Recommendation:** 4

**Summary:**

This paper addresses a fundamental limitation in Autoregressive Models (ARMs)—the assumption of a fixed generation order, which may not be optimal for complex data types like graphs. The authors introduce Learning-Order Autoregressive Models (LO-ARMs), a novel approach where the model learns a context-dependent order for sequential data generation rather than relying on a predefined or random order. This paper presents a major advancement in autoregressive modeling, showing that learning an optimal generation order improves sample efficiency and generation quality.

**Claims And Evidence:**

The majority of the claims in the paper are supported by well-structured evaluations.

**Essential References Not Discussed:**

No

**Experimental Designs Or Analyses:**

All good

**Methods And Evaluation Criteria:**

The proposed methods and evaluation criteria mostly align well with the problem of learning dynamic generation orderings in autoregressive models, especially for molecular graph generation. However, some aspects of the methodology and evaluation could be further discussed.

The paper states, "LO-ARM generalizes across different molecular graphs" (p. 7), but only evaluates on QM9 and ZINC250K, both of which contain relatively simple organic molecules. The datasets such as larger bioactive molecules (ChEMBL) could be helpful to check the ability of the model on the complex molecular graphs.

**Other Comments Or Suggestions:**

The bold result in table is not clear, it looks like authors are bold the best result on the other method and the on their proposed method in table 1. But in table 2, the best result on proposed method is not bolded. It would be great to keep them consistent.

Also please considering to cite the survey papers for molecule/graph generation.

**Other Strengths And Weaknesses:**

See above

**Questions For Authors:**

No.

**Relation To Broader Scientific Literature:**

The key contributions of this paper build upon and extend several established areas in machine learning, particularly autoregressive models (ARMs), variational inference, graph generation, and molecular generative modeling.

**Theoretical Claims:**

No issues.

---

> ### Author Rebuttal · Authors · 2025-04-01
>
> We sincerely appreciate your positive feedback on the quality of our work. We now address each of your concerns as below.
>
> ## 1. Preliminary results on larger bioactive molecule dataset (ChEMBL)
>
> To address your concern regarding the scalability of LO-ARM to larger datasets, we include a preliminary result on the ChEMBL dataset (also known as the GuacaMol benchmark [1]) without hyperparameter tuning. Specifically, we preprocessed this dataset using the utility provided in [2], a method consistently employed in our benchmarks [2, 3]. We have summarized the statistics of the three datasets used in our evaluation—namely QM9, ZINC250k, and ChEMBL/GuacaMol—as presented below. We believe this inclusion provides a more comprehensive assessment of LO-ARM's capabilities.
>
>
>
> |          | #samples | #nodes           | #node types | Input dims |
> |----------|----------|------------------|-------------|------------|
> | QM9      | 130k     | 1 <= \|V\| <= 9  | 4           | 90        |
> | ZINC250k | 250k     | 6 <= \|V\| <= 38 | 9           | 1482        |
> | GuacaMol | 1.2M     | 2 <= \|V\| <= 62 | 12          | 3906       |
>
>
> Our preliminary results show that LO-ARM still exceeds or matches performance of the current state-of-the-art-models.
>
> |               | Input data | Validity | Uniqueness | Novelty   | Raw FCD  |
> |----------|--------|--------|---------|---------|--------|
> | LSTM          | SMILES     | **95.9** | **100.0**  | 91.2      | **0.46** |
> | MCTS          | Graph      | 92.9     | 95.5       | 100.0     | 21.00    |
> | DiGress       | Graph      | 85.2     | 100.0      | 99.9      | 1.92     |
> | LO-ARM (ours) | Graph      | *94.2*   | **100.0**  | **100.0** | 3.73     |
>
> To ensure consistency in our evaluation across the QM9 and ZINC250k datasets (as is common in the literature), we have converted the reported FCD scores from prior work, which are often presented on an exponential scale, back to their raw FCD values.
>
> Specifically, the LO-ARM model trained on the GuacaMol dataset exhibits a preference for an atom-first generation order. This means it tends to generate atoms first, followed by the real bonds connecting them, and lastly, it fills in the non-existent or "imaginary" bonds. We illustrate an example of this generation process in the accompanying figure [4]. This learned ordering contrasts with the edge-first ordering observed in models trained on the ZINC250k and QM9 datasets. This difference likely arises because generating node-related dimensions first is potentially simpler for the GuacaMol samples, given that the number of edge dimensions increases quadratically with the number of nodes.
>
> Our preliminary findings indicate that LO-ARM surpasses other graph-based methods in terms of validity and uniqueness, achieving performance levels close to the state-of-the-art SMILES-based approaches. However, we have observed a small gap in FCD compared to DiGress. We hypothesize that this difference arises because for datasets with higher dimensionality, a generation strategy operating at a coarser granularity (e.g., dimension blocks) than individual dimensions might be more effective. For example, tokenizing molecules into fragments, as discussed in our Discussion section, could potentially improve performance. We defer the task of integrating domain-specific tokenization into LO-ARM for future investigation.
>
> ## 2. Some aspects of the methodology and evaluation could be further discussed.
> In addition to these new results of GuacaMol experiments, we have provided an in-depth analysis on training stability with REINFORCE in the thread of Reviewer Stjb.
>
> ## 3. Improve the presentation of the main tables and enrich related work
> Thank you for your thorough review and your advice, and we will refine the main tables and reference the surveys of graph generation in the final version.
>
> We hope that our responses have addressed your primary oncerns regarding the scalability of our algorithm. If so, we respectfully request that the reviewer reconsider their decision in light of our responses and update their recommendation score accordingly. We remain available and eager to address any further concerns the reviewer may have.
>
> [1] https://arxiv.org/abs/1811.09621
>
> [2] https://arxiv.org/abs/2209.14734
>
> [3] https://arxiv.org/abs/2406.04843
>
> [4] Generation trajectory of GuacaMol sample: https://drive.google.com/file/d/1a5HU4FZ98bqS_9JFK4Eb60QfhXiksaue/view?usp=drive_link

---

> > ### Comment · Reviewer_Pa4s · 2025-04-07
> >
> > Thanks authors for adding the dataset and additional experiments. I have no further concerns.

---

### Official Review · Reviewer_Stjb · 2025-03-13

**Overall Recommendation:** 3

**Summary:**

This paper proposes a method to learn an optimal generation order for autoregressive models in data domains that do not possess a natural canonical ordering (e.g., graphs or images). By framing the ordering itself as a latent variable with a dynamic, learnable distribution (the “order-policy”), the authors unify the ideas of AO-ARMs with variational inference. They demonstrate that a variational bound on the log-likelihood can be optimized by sampling permutations from a “posterior” distribution and matching them to the model’s own “order-policy”. Empirically, the method achieves competitive results on two molecular generation datasets, QM9 and ZINC250k. The paper’s thorough ablations highlight design trade-offs for parameterizing the order-policy, and the final approach shows consistent improvements over both uniform and biased AO-ARMs.

**Claims And Evidence:**

The main claims of the paper are:
1. By introducing a trainable policy over generation order, the autoregressive mode; avoids the limitations of either a single fixed order or a uniformly random permutation of dimensions.

2. The authors propose a variational approach, using REINFORCE with a leave-one-out baseline to reduce gradient variance. They argue that this is practical enough to handle the large combinatorial space of permutations.

These claims are generally supported by the quantitative comparisons.

My concerns are 1. the convergence of REINFORCE-based training is not thoroughly studied in the analysis session; 2. the performance gains on the two datasets (QM9 and ZINC250k) are very modest as the baselines have already achieved very high results.

**Essential References Not Discussed:**

None.

**Experimental Designs Or Analyses:**

The experiments on molecular generation are thorough, with ablations revealing the roles of shared vs. separate neural networks for the order-policy and the choice of top-p sampling strategies. The authors also visualize generation paths for example molecules, giving a nuanced look at how the dynamic ordering policy emerges in practice.

However, concerns remain about REINFORCE’s notorious variance. While the authors do implement a leave-one-out method to reduce variance, readers may question the stability of training at larger scales or on smaller datasets with fewer training points (like FreeSolv or Lipophilicity). An in-depth discussion or demonstration of how many epochs/hyperparameter adjustments were needed would be welcome.

**Methods And Evaluation Criteria:**

The authors follow common practice in generative modeling by: evaluating sample quality on the standard Validity/Uniqueness metrics for molecules, and measuring distributional similarity with FCD, a recognized measure for drug-like molecules.

Although the results are strong, the paper focuses primarily on mid-scale tasks (QM9, ZINC). It would be beneficial to demonstrate that the approach still offers consistent improvements on small-scale benchmarks [1,2,3], as the baseline results on these benchmarks are less optimal compared to QM9 and ZINC.

[1] FreeSolv: a database of experimental and calculated hydration free energies.
[2] In silico evaluation of logD7.4 and comparison with other prediction methods.
[3] The Harvard organic photovoltaic dataset.

**Other Comments Or Suggestions:**

None.

**Other Strengths And Weaknesses:**

None.

**Questions For Authors:**

1. Have you measured or visualized the gradient variance over the course of training? Are there scenarios (like larger, more complex data) where you suspect REINFORCE might fail to converge easily?

2. Do you have preliminary results on smaller tasks (e.g., FreeSolv, Lipophilicity, HOPV) that confirm the order-policy still yields an advantage?

**Relation To Broader Scientific Literature:**

The paper situates its method well with respect to AO-ARMs, masked discrete diffusion, and existing graph-generation methods. It clarifies that past approaches either fixed or uniformly randomized the ordering.

**Theoretical Claims:**

The derivation of the method as a variational ARM with a learned ordering distribution is a clean extension to the standard AO-ARM framework. The theoretical foundation seems sound: the ordering z serves a hidden latent variable for the distribution of data x, so the standard variantional inference can play a role here for the optimization of joint probability distribution.

---

> ### Author Rebuttal · Authors · 2025-04-01
>
> Thank you for the positive feedback! We are glad that you find our theoretical results sound, our improvement consistent and our ablations thorough. We address your concerns as below, especially your concern on training stability with REINFORCE.
>
> ### 1. Performance gains on QM9 and ZINC250k
> We believe our improvements on QM9 and ZINC250k are substantial. Although baselines score well on validity/uniqueness, our results indicated that there is still large room for improvement in distributional similarity metrics such as FCD — highlighted by Reviewer Stjb as a recognized measure for drug-like molecules. Our LO-ARMs model achieved new state-of-the-art FCD scores, reducing them to 0.240 (QM9) and 3.229 (ZINC250k) from prior bests of 0.441 and 13.21, respectively.
>
> ### 2. Smaller and larger scale benchmarks (e.g., FreeSolv, Lipophilicity, HOPV)
> Thank you for your suggestion, and we will certainly include the results on smaller benchmarks in the final version of our work. Given the time constraints of this rebuttal period, we made a strategic decision to prioritize the GuacaMol experiment, which is a much larger dataset than ZINC250k, to maximize the efficiency of our response. This is because we anticipate that training stability might pose a greater challenge for larger datasets compared to smaller ones, as their training processes tend to exhibit more stochasticity. Furthermore, the scalability of LO-ARM is a concern shared by several reviewers. The detailed results on GuacaMol dataset are presented in the response to Reviewer Pa4s. In short, our preliminary results show that LO-ARM learns an order-first generation ordering [1] and still exceeds or matches performance of the current state-of-the-art-models.
>
> ### 3. Analysis of variance and convergence of REINFORCE-based training
> To provide a clearer understanding of the variance and convergence of REINFORCE and the effectiveness of our variance reduction method, we have visualized the following quantities during training course of the GuacaMol experiments [2]:
>
> * Negative Evidence Lower Bound (ELBO)
> * Maximum and minimum q-logits from the variational order policy network.
>
> Specifically, we conducted an ablation study on the learning rate in two experiments, keeping all other settings constant. Given that the gradient variance primarily arises from the stochasticity of the variational order policy, the maximum and minimum q-logits can reflect this variance throughout the training.
>
> As seen in the plot, most of the time the RLOO variance reduction keeps the optimization smooth -- still occasionally training instability is observed with a large learning rate (2e-5), indicated by loss spikes and discontinuities in the maximum q-logits. Fortunately, reducing the learning rate to 1.5e-5 effectively stabilizes training, as demonstrated by the green curves.
>
> We hope that our responses have addressed your concerns and questions. If so, we respectfully ask that you reconsider your decision based on our responses and update your recommendation score accordingly. We are also eager to address any further concerns you may have.
>
> [1] Generation trajectory of GuacaMol sample: https://drive.google.com/file/d/1a5HU4FZ98bqS_9JFK4Eb60QfhXiksaue/view?usp=drive_link
>
> [2] Variance visualization: https://drive.google.com/file/d/19b2wQocvLJc82bAm6eUG5qbAcmVAra0V/view?usp=drive_link

---

### Official Review · Reviewer_H7W8 · 2025-03-17

**Overall Recommendation:** 3

**Summary:**

This paper propose a new generative modeling framwork named  Learning-Order Autoregressive Models. The core of this framework is to extend the traditional ARs to learning a dynamic order of sampling. Speicifically train a order polocy to determine the order. To train such model, the authers use a  variational lower bound on the exact log-likelihood , and optimize via stochastic gradient estimation. It achieved quite good results in  QM9 and Zink250K.

## Update after rebuttal
I appreciate the authors' thorough response to the concerns raised. I have no further questions and will maintain my original score as positive unchanged. Thank you for your efforts.

**Claims And Evidence:**

I agree that the authors are dealing with a important issue in  graph autoregressive generation as it is hard to determine a order for sampling. I think use a order-policy to sample the order seems reasonable. The claim of LO-ARMs Learn a Meaningful and Consistent Order for Generation is acceptable.
While LO-ARMs Can Generalize to Other High-Dimensional Data (Images, Graphs) shoule be given more evidence. The MNIST experiment is more of a toy example rather than rigorous proof that the method generalizes to all high-dimensional data, and quantitative metrics (e.g., likelihood scores, FID, etc.) should be  provided for image generation, making it easy to measure actual performance compared to baselines.

**Essential References Not Discussed:**

Related works are discussed properly.

**Experimental Designs Or Analyses:**

The experiment settings are from the existing related work, and the authors use the same settings, which are reasonable

**Methods And Evaluation Criteria:**

Overall the the method proposed is suitable for the problem . The HyperEdge enhanced EGNN and the docking scorefeatures are reasonable. The evaluations are on the commonly used unconditional molecule graph generation.

**Other Comments Or Suggestions:**

No

**Other Strengths And Weaknesses:**

Strengths
The paper presents a novel approach to learning orderings in autoregressive models, which is a non-trivial extension of traditional ARMs. The paper provides a well-structured training algorithm and a clear sampling procedure, making the method well-grounded
Weaknesses
While LO-ARMs are tested on MNIST as a toy example,  i think this is not sufficient to claim general applicability to high-dimensional data like images, more examples of benchmark in image generation would be better.
Would be better if a pipeline illstration figure is provided. Currently only a case study figure shown in the paper main part.

**Questions For Authors:**

See the Strengths And Weaknesses part

**Relation To Broader Scientific Literature:**

Appling the order policy to other generation modalitys like 3d point cloud, pixels, voxels maybe interesting.

**Theoretical Claims:**

There are no theoretical claims in this paper

---

> ### Author Rebuttal · Authors · 2025-04-01
>
> Thank you for the time you’ve taken to review our work and for the positive and constructive feedback! We are glad that you found the problem we dealt with important and our approach novel and well-grounded. In response to the weaknesses and questions:
>
> ### 1. Generalization to other high-dimensional data (images, graphs)
> We would like to clarify that the focus of this work is on designing the order learning mechanism in ARMs and its application to molecular graph generation as the main testbed. We chose this task because we believe it is more suitable – As the reviewer has pointed out, an important issue in graph autoregressive generation is the difficulty to determine an order for sampling. Indeed, our learning-order ARM was able to discover an autoregressive order that outperforms fixed or random order used in prior graph generation work. Our focus is not on images – we included MNIST only as a sanity check to confirm the model can learn a meaningful order that distinguishes between the digits and the background. While there are no practical constraints for applying LO-ARMs to higher dimensional natural images – it is unclear whether there is a similarly meaningful and interpretable ordering to learn there and therefore we decided not to pursue it in this work.
>
> To demonstrate the scalability of our algorithm to higher dimensional graph datasets, we conduct an additional experiment on the GuacaMol dataset, which is a larger molecule dataset with 3906 input dimensions (which is larger than 1482 of ZINC250k and 90 of QM9). We have provided the comparison of the three molecule datasets in the discussion with Reviewer Pa4s. We believe that these three datasets can provide a comprehensive evaluation matrix to support our claims.
>
> Performance-wise, our preliminary results show that the order-policy still yields an advantage, exceeding or matching the SOTA performance, as you can see from the details we have provided in the discussion thread with Reviewer Pa4s.
>
> ### 2. Application to 3d point cloud, pixels, voxels
> Thank you for the suggestion. We are interested in exploring these modalities in future work.
>
> ### 3. Pipeline illustration figure
> We will follow the reviewer’s suggestion to add an illustration figure in the final version.
>
> We hope that our responses have addressed your concerns and questions. If so, we would kindly ask the reviewer to reconsider the decision in light of our responses and update their score accordingly. We are also eager to address any additional concerns the reviewer may have.

---

### Decision · Program_Chairs · 2025-05-01

**Decision:**

Accept (poster)

**Comment:**

The paper introduces an auto-regressive model (ARM) that generates images and graphs using a probabilistic ordering inferred from data. This is achieved by using a trainable probability distribution that dynamically decides the sampling order of the data dimensions, which is optimized using stochastic gradient estimation the variation lower bound of the exact log-likelihood. Experimentally, the paper shows good results on molecular generation (QM9 and ZINC250k benchmarks).

This paper was well-received by the reviewers, in particular the clarity of exposition and the motivations behind the paper, the novelty and elegance of the approach, and the satisfactorily experimental part. The rebuttal phase seems to have addressed most concerns, and while some still remain (see the response by reviewer u8KG which points up a higher number of sampling steps with respect to e.g. SMILES generators) they seem minor.

I recommend acceptance.